# Evolutionary dynamics of transposable elements in bdelloid rotifers

Reuben W Nowell[1,2]*, Christopher G Wilson[1,2], Pedro Almeida[2,3], Philipp H Schiffer[4], Diego Fontaneto[5], Lutz Becks[6,7], Fernando Rodriguez[8], Irina R Arkhipova[8], Timothy G Barraclough[1,2]*

[1]Department of Zoology, University of Oxford, Oxford, United Kingdom; [2]Department of Life Sciences, Imperial College London, Silwood Park Campus, Ascot, Berkshire, United Kingdom; [3]Division of Biosciences, University College London, London, United Kingdom; [4]Institute of Zoology, Section Developmental Biology, University of Cologne, Köln, Wormlab, Germany; [5]National Research Council of Italy, Water Research Institute, Verbania Pallanza, Italy; [6]Community Dynamics Group, Department of Evolutionary Ecology, Max Planck Institute for Evolutionary Biology, Plön, Germany; [7]Aquatic Ecology and Evolution, University of Konstanz, Konstanz, Germany; [8]Josephine Bay Paul Center for Comparative Molecular Biology and Evolution, Marine Biological Laboratory, Woods Hole, MA, United States

*For correspondence:
reuben.nowell@zoo.ox.ac.uk (RWN);
tim.barraclough@zoo.ox.ac.uk (TGB)

**Competing interests:** The authors declare that no competing interests exist.

**Abstract** Transposable elements (TEs) are selfish genomic parasites whose ability to spread autonomously is facilitated by sexual reproduction in their hosts. If hosts become obligately asexual, TE frequencies and dynamics are predicted to change dramatically, but the long-term outcome is unclear. Here, we test current theory using whole-genome sequence data from eight species of bdelloid rotifers, a class of invertebrates in which males are thus far unknown. Contrary to expectations, we find a variety of active TEs in bdelloid genomes, at an overall frequency within the range seen in sexual species. We find no evidence that TEs are spread by cryptic recombination or restrained by unusual DNA repair mechanisms. Instead, we find that that TE content evolves relatively slowly in bdelloids and that gene families involved in RNAi-mediated TE suppression have undergone significant expansion, which might mitigate the deleterious effects of active TEs and compensate for the consequences of long-term asexuality.

## Introduction

Transposable elements (TEs) are repeated sequences of DNA that can mobilise and replicate themselves within genomes (*Charlesworth and Charlesworth, 1983*; *Hickey, 1982*; *Orgel and Crick, 1980*). TEs are divided into two major categories: class I retrotransposons, which use a 'copy-and-paste' replication mechanism via a reverse-transcribed RNA intermediate, and class II DNA transposons, which use 'cut-and-paste' replication with a DNA intermediate. Both classes are ancient and diverse—retrotransposons are found in some bacteria and nearly all eukaryotes, while DNA transposons are found across the tree of life (*Doolittle et al., 1989*; *Eickbush and Malik, 2002*; *Robertson, 2002*). Although TE replications are occasionally beneficial (*Capy et al., 2000*), the vast majority are deleterious for the host (*Bourgeois and Boissinot, 2019*; *Burt and Trivers, 2009*). Costs include insertional mutations that disrupt genes (*Finnegan, 1992*), cellular costs of replicating and expressing excess DNA (*Nuzhdin, 1999*), and increased risk of chromosomal abnormalities due to ectopic recombination between homologous TE sequences interspersed through the genome (*Langley et al., 1988*; *Montgomery et al., 1987*). Despite this, TEs can accumulate to large

numbers by replicating autonomously as selfish elements within genomes—for example, TEs comprise 46% of the human genome, including over half a million class I long interspersed elements (LINEs) from the *L1* subfamily (*Craig, 2015*; *International Human Genome Sequencing Consortium et al., 2001*). TE numbers can vary greatly, however, even between closely related species. In vertebrates, for example, TE proportion spans an order of magnitude, from below 6% to over 50% of the genome (*Chalopin et al., 2015*), with similarly large variation observed in other groups such as arthropods (*Petersen et al., 2019*), nematodes (*Szitenberg et al., 2016*), fungi (*Castanera et al., 2016*), and plants (*Bennetzen and Wang, 2014*; *Feschotte et al., 2002*). Explaining this variation is vital to understanding the mechanisms affecting TE spread and control.

Sexual reproduction has long been thought to play a major role in TE dynamics within eukaryotes. On the one hand, sexual reproduction and outcrossing decouples the fate of TEs from other host genes, allowing them to jump into new genomic backgrounds and behave as selfish genomic parasites (*Doolittle and Sapienza, 1980*; *Hickey, 1982*; *Orgel and Crick, 1980*). On the other hand, sex enables the efficient removal of deleterious insertions from populations through recombination and segregation (*Charlesworth and Langley, 1989*; *Schaack et al., 2010a*; *Wright and Finnegan, 2001*). The risk of chromosome abnormalities arising from ectopic recombination, arguably the main cost of high TE loads in eukaryotes (*Bourgeois and Boissinot, 2019*; *Petrov et al., 2003*), also occurs during chromosome pairing at meiosis. Sex therefore plays opposing roles—it permits spread and selfish behaviour of TEs, and yet it facilitates and strengthens selection against high loads. Variation in TE content among taxa might thus result from shifts in the balance of these different opposing forces.

By this logic, the loss of sexual reproduction should affect TE dynamics dramatically. Since asexual lineages generally arise from sexual species, it is likely that they initially harbour many active TEs (*Charlesworth and Langley, 1986*; *Dolgin and Charlesworth, 2006*; *Robertson, 2002*; *Schaack et al., 2010b*). All else being equal, the loss of sex will limit the ability of selection to remove deleterious insertions from a fully linked host genome, and so the load of TEs should accumulate. At the same time, the fate of TEs is immediately coupled to that of the host genome, resulting in intensified selection for inactivation, excision, or domestication of the elements (*Bast et al., 2019*; *Dolgin and Charlesworth, 2006*; *Fujita et al., 2020*; *Hickey, 1982*; *Wright and Finnegan, 2001*). The genomes of asexual lineages whose TEs continued to replicate unchecked would become overrun, potentially leading to the extinction of both the lineage and the TEs themselves.

Models of the population genetics of vertically transmitted TEs in asexuals therefore predict one of two outcomes: either TEs accumulate within lineages faster than they can be removed, overrunning each lineage in turn and driving the population extinct, or, conversely, TE removal outweighs proliferation and the population eventually purges itself entirely of deleterious TEs (*Boutin et al., 2012*; *Dolgin and Charlesworth, 2006*; *Startek et al., 2013*). These predictions should apply particularly to TEs from the LINE-like group of class I retroelements, as these are thought to be almost exclusively vertically transmitted owing to the instability of the extrachromosomal RNA intermediate (*Eickbush and Malik, 2002*; *Peccoud et al., 2017*; *Robertson, 2002*; *Schaack et al., 2010b*; *Silva et al., 2004*). In contrast, it is thought that some class II DNA elements are maintained by horizontal transfer between species (*Robertson, 2002*; *Schaack et al., 2010b*) or by having beneficial effects (as in bacteria; *Basten and Moody, 1991*; *Edwards and Brookfield, 2003*). Class II TEs might therefore be less dependent on sex, on average, and consequently less affected by its loss. These predictions are difficult to test empirically, however, because it is expected to take millions of generations for an asexual population to either eliminate TEs or go extinct (*Dolgin and Charlesworth, 2006*), too long to observe directly and beyond the lifespan of most asexual lineages (*Jaron et al., 2020*; *Neiman et al., 2009*).

Here, we test these ideas in a well-known group of asexual animals, the bdelloid rotifers. These microscopic invertebrates appear to have reproduced without males or meiosis for tens of millions of years, diversifying into hundreds of species within limno-terrestrial and freshwater habitats globally (*Mark Welch et al., 2009*; *Robeson et al., 2011*). Bdelloids sampled from nature (and those reared in the laboratory) consist entirely of parthenogenetic females, and neither males nor hermaphrodites are described for any species despite centuries of close observation by naturalists (*Donner, 1965*; *Hudson and Gosse, 1886*; *Mark Welch et al., 2009*). Genetic and genomic evidence for their proposed ancient and obligate asexuality remains uncertain, however. Initial evidence of long-term asexuality (*Flot et al., 2013*; *Mark Welch and Meselson, 2000*) has been refuted by

later studies or confounded by alternative explanations (*Mark Welch et al., 2008*; *Nowell et al., 2018*; *Simion et al., 2020*). Some recent studies have proposed alternative modes of inter-individual genetic exchange, but these suggestions would require exotic mechanisms unknown in other animals (*Flot et al., 2013*; *Signorovitch et al., 2015*), or rates of sex that are difficult to reconcile with the lack of observed males (*Vakhrusheva et al., 2020*). While the precise genetic system in bdelloids remains an open question, nonetheless they provide a unique test-case for models of TE evolution when conventional sex is absent or strikingly rare.

Initial PCR-based surveys of five bdelloid genomes found no evidence of class I retrotransposons from either the long terminal repeat (LTR) or LINE superfamilies, but did reveal a diverse array of class II DNA transposons, mostly at low copy number (*Arkhipova and Meselson, 2000*). This finding was interpreted in the light of the predicted effects of the loss of sex outlined above. Specifically, the apparent lack of class I retrotransposons in bdelloids contrasted sharply with their near ubiquity in other taxa, and at the time appeared consistent with the view that long-term asexual evolution in bdelloids had caused the loss of parasitic elements that depended on sexual transmission (*Arkhipova and Meselson, 2005*; *Arkhipova and Meselson, 2000*; *Dolgin and Charlesworth, 2006*; *Wright and Finnegan, 2001*). In contrast, the class II TEs were hypothesised to be maintained by horizontal gene transfer (HGT), a process that occurs to an unusual degree in bdelloid rotifers from non-metazoan sources (*Boschetti et al., 2012*; *Eyres et al., 2015*; *Flot et al., 2013*; *Gladyshev et al., 2008*; *Nowell et al., 2018*), and hence able to persist despite the loss of sex.

Another unusual aspect of bdelloid physiology was suggested to contribute to their seemingly low TE complement. In most bdelloid species (but not all), individuals can survive complete desiccation at any life stage via a process called anhydrobiosis ('life without water'). Desiccation causes double-strand breakages (DSBs) in DNA, but bdelloids are able to repair these and recover to an uncommon degree (*Gladyshev and Meselson, 2008*; *Hespeels et al., 2014*; *Ricci, 1998*). It was proposed that anhydrobiosis might influence TE evolution in two ways (*Arkhipova and Meselson, 2005*; *Dolgin and Charlesworth, 2006*; *Gladyshev and Arkhipova, 2010a*). First, DSB repair could provide a mechanism for intragenomic recombination that aids TE removal, either via gene conversion from a homologous chromosome lacking the TE insertion, or excision of mis-paired regions. Second, the pairing of homologous chromosomes, if required during DSB repair, could provide a context for ongoing selection against chromosomal abnormalities caused by ectopic recombination. In either case, anhydrobiosis would decrease the number of TEs, potentially helping to explain the low overall TE content encoded in bdelloid genomes.

These early ideas were transformed by more detailed studies of the model bdelloid species *Adineta vaga*, which used refined methods and genome-scale data to discover a variety of retrotransposon families. These include an endonuclease-deficient *Penelope*-like element (PLE) designated *Athena* (*Arkhipova et al., 2003*; *Gladyshev and Arkhipova, 2007*), which is itself incorporated within much larger and highly unusual retroelements called *Terminons* (*Arkhipova et al., 2017*), another PLE that has retained its endonuclease (*Arkhipova et al., 2013*), LTR retrotransposons (*Juno*, *Vesta*, *TelKA*, and *Mag*) (*Gladyshev et al., 2007*; *Rodriguez et al., 2017*), and LINE-like retrotransposons (*R4*, *R9*, *Hebe*, *RTE*, *Tx1*, and *Soliton*) (*Flot et al., 2013*; *Gladyshev and Arkhipova, 2010b*; *Gladyshev and Arkhipova, 2009*). In total, TEs accounted for 2.2% of the 217 Mb genome (~4.8 Mb; *Flot et al., 2013*), rising to ~4% on inclusion of the recently discovered giant *Terminon* elements (*Arkhipova et al., 2017*). Whole genome sequencing also provided clues into molecular mechanisms that might limit TE activity in bdelloids. Specifically, multiple copies of genes involved in RNAi pathways were discovered in *A. vaga* (*Flot et al., 2013*), which play a role in suppressing TE activity. It was hypothesised that these gene families might have undergone significant expansion, but this was not formally tested.

While numerous investigations confirmed the presence of both class I and class II TEs in bdelloids, the effects of asexuality and anhydrobiosis on TE evolution remain open questions. Specifically, the different hypotheses can be evaluated by pinpointing the location and direction of any shifts in TE profiles on a phylogenetic tree that incorporates bdelloids and their relatives. All bdelloid rotifers share the same system of reproduction that differs from their nearest relatives, whereas not all bdelloid lineages survive desiccation (*Ricci, 1998*). The asexuality hypothesis therefore predicts a major shift in TE profiles along the stem branch for bdelloids as a whole, especially a decline in the abundance of class I TEs expected to be most affected by the loss of sex, whereas the desiccation hypothesis predicts further shifts within the bdelloid clade correlated with desiccation tolerance.

Alternatively, there might be no significant shift in the TE profile of bdelloid rotifers compared to background rates of evolution in TE content across animals. Such a finding might indicate that one or more assumptions of the theory of TE evolution in asexuals are not met in bdelloid rotifers.

Here, we test these predictions by quantifying TE evolution across 42 rotifer genomes belonging to 15 taxonomic species. Our sample includes both desiccating and nondesiccating bdelloids, and eight monogonont rotifers (*Blommaert et al., 2019*; *Han et al., 2019*; *Kim et al., 2018*), a separate class that alternates sexual and asexual reproduction and cannot survive desiccation as adults. Further phylogenetic context is provided by published genomes from an acanthocephalan (*Mauer et al., 2020*), an obligately sexual clade now classified with rotifers in the Phylum Syndermata, and a range of other animal phyla, including additional comparisons between desiccating and nondesiccating taxa. We use phylogenetic models to test the predictions of the asexuality and desiccation hypotheses outlined above and perform further analyses to explore the assumptions behind these theories as applied to bdelloids. Specifically, we investigate whether TEs remain recently active, whether TE polymorphism within populations is indeed consistent with asexual inheritance, whether bdelloid TEs experience the same selective constraints as in other animals, and whether TE defence pathways have expanded in bdelloids as previously proposed.

## Results and discussion

### High-quality comparative genomics data for bdelloid rotifers

To quantify variation in repeat content within and between bdelloid species, we generated de novo whole-genome assemblies for 31 rotifer samples encompassing nine species (*Figure 1A*, *Table 1*, *Table 1—source data 1*). Three of these assemblies were generated using 10x Genomics linked-read data from clonal populations grown from single wild-caught individuals (for *Adineta steineri*, *Rotaria sordida*, and *Rotaria* sp. 'Silwood-1'), while 26 are from Illumina libraries of DNA extracted directly from single wild-caught individuals. In order to capture as many potential repeats as possible, we generated two assemblies for each Illumina sample: a 'reference' assembly, with a focus on quality and contiguity, and a 'maximum haplotype' (maxhap) assembly that included small or highly similar contigs that might be derived from recent TE duplications or other sources of copy number variation, at the expense of contiguity.

Reference genomes showed an expected scaffold size (AU, see Materials and methods) ranging from 21.1 kb (*Didymodactylos carnosus*) to 702.3 kb (*R.* sp. 'Silwood-1') and BUSCO scores that indicated 89–98% of 303 core eukaryote genes were completely recovered, increasing to 96–99% if fragmented BUSCO copies are included (*Table 1*). General genome characteristics such as genome size (assembly span), the proportion of G + C nucleotides (GC%), the number of coding genes (CDS), and the level of intragenomic homologous divergence (number of SNPs identified within CDS) were within the range expected from previous analyses of bdelloid genomes (*Flot et al., 2013*; *Nowell et al., 2018*) (*Figure 1B–C*, *Table 1*, *Figure 1—figure supplement 1*). Intragenomic collinearity and synonymous divergence of coding regions in the *A. steineri*, *R. sordida*, and *R.* sp. 'Silwood-1' 10x Genomics diploid assemblies revealed the characteristic signature of degenerate tetraploidy that has been found in all bdelloid species examined to date (*Figure 1D*).

Compared to the reference set, maxhap assemblies generally showed increased span (mean increase = 17.9 Mb ± 21.5 standard deviation [SD]) and were substantially more fragmented, as expected (*Table 1—source data 2*). Nonetheless, BUSCO completeness scores remained high, with 76–98% of genes completely recovered (increasing to 95–98% if fragmented copies are included), indicating that the majority of core genes are successfully captured (*Table 1—source data 2*). The BUSCO duplication metric ('D') does not increase greatly between reference and maxhap assemblies, which shows that the additional sequences retained in the maxhap assemblies do not contain complete extra copies of core genes. Thus, the maxhap assemblies are not fully haplotype-resolved representations of the genome, except in the case of the three 10x assemblies.

To these new data, we added published genomes for four bdelloids (*A. vaga*, *Adineta ricciae*, *Rotaria magnacalcarata* and *Rotaria macrura*) (*Flot et al., 2013*; *Nowell et al., 2018*) and seven monogononts: one from the *Brachionus calyciflorus* species complex (*Kim et al., 2018*) and six from four species of the *Brachionus plicatilis* species complex, namely *B. asplanchnoidis*, *B. plicatilis* sensu stricto (HYR1), *B. rotundiformis*, and *B.* sp. 'Tiscar' (*Blommaert et al., 2019*; *Han et al., 2019*),

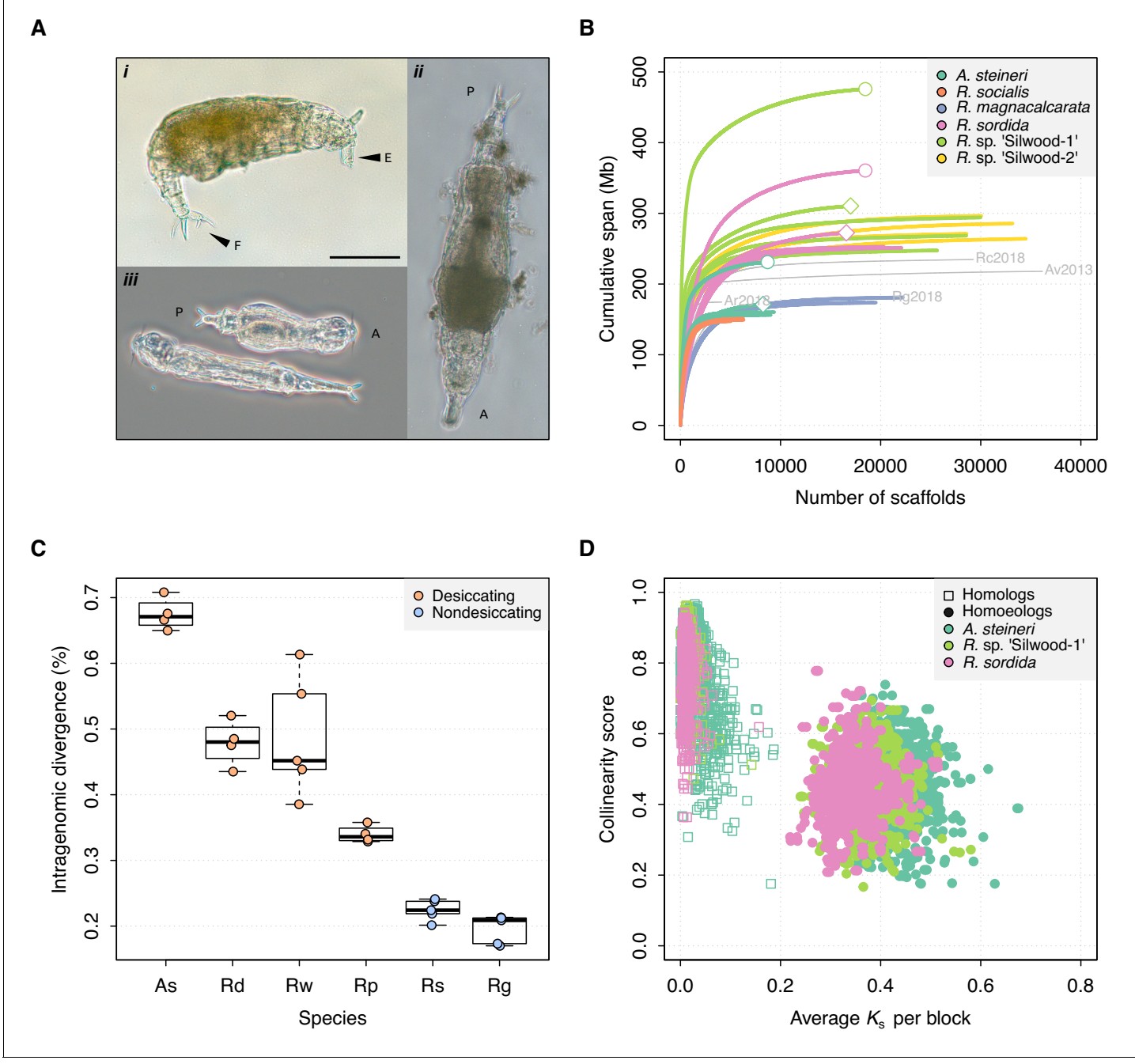

**Figure 1.** Genome properties of sequenced rotifers. (**A**) Bdelloid rotifer morphology; scale bar indicates 100 μm. (*i*) Individual from an undescribed species of *Rotaria* (*R.* sp. 'Silwood-1'), showing eyes (E) and foot (F) with two spurs and three toes. (*ii*) Further image of *R.* sp. 'Silwood-1' with anterior–posterior (A–P) axis marked. (*iii*) Two individuals of *A. steineri* in phase contrast illumination. (**B**) Cumulative assembly span for six bdelloid species with population genomics data (*n* > 2). 10x Genomics haploid ('pseudohap') and diploid ('megabubbles') assemblies for *A. steineri*, *R. sordida* and *R.* sp. 'Silwood-1' are indicated with diamond and circle symbols, respectively. The four previously published genomes for *A. vaga* ('Av2013', GenBank accession GCA_000513175.1) and *A. ricciae* ('Ar2018', GCA_900240375.1), *R. macrura* ('Rc2018', GCA_900239685.1) and *R. magnacalcarata* ('Rg2018', GCA_900239745.1) are indicated in grey, for comparison. (**C**) Intragenomic homologous divergence (i.e. heterozygosity), measured as the number of SNPs detected in coding regions (CDS). Boxplots show the median (band), interquartile range (box) and minimum/maximum values (whiskers). Underlying data are shown as jittered points. Desiccation-tolerant species are in orange, intolerant species in blue. Species abbreviations: **As**, *A. steineri*; **Rd**, *R. sordida*; **Rw**, *R.* sp. 'Silwood-1'; **Rp**, *Rotaria* sp. 'Silwood-2'; **Rg**, *R. magnacalcarata*; **Rs**, *R. socialis*. (**D**) Genome structure in *A. steineri*, *R. sordida* and *R.* sp. 'Silwood-1' haplotype-resolved ('megabubbles') assemblies. Each point represents a collinear block of genes, plotted by average pairwise synonymous ($K_s$, *X*-axis) and collinearity score (see Materials and methods and S1 note) on the *Y*-axis. Separation into two distinct clusters

*Figure 1 continued on next page*

*Figure 1 continued*

representing homologous (squares) and homoeologous (circles) relationships among gene copies is consistent with ancestral tetraploidy, with homoeologous copies derived from a putative ancient genome duplication.

The online version of this article includes the following figure supplement(s) for figure 1:

**Figure supplement 1.** Genome characteristics for bdelloid samples.

yielding a total of 42 rotifer genomes. Of these, 11 samples belong to nondesiccating bdelloid species (five individuals each from *R. magnacalcarata* and *Rotaria socialis*, and the previously published genome of *R. macrura*). The inclusion of *Didymodactylos carnosus* helps broaden the bdelloid sampling because it ensures that trees include the deepest molecular divergences known within the class to date. Nonetheless, while our study includes all publicly available genome sequences for both bdelloids and monogononts available at the time of analysis, both groups comprise a large diversity of genera, much of which remains unsampled.

Our final representative of the phylum Syndermata is the recently published genome of the obligately sexual acanthocephalan (thorny-headed worm) *Pomphorhynchus laevis* (*Mauer et al., 2020*). Although previously classified as a separate phylum, increasing evidence suggests that acanthocephalans may be the closest relatives to the Class Bdelloidea (*Laumer et al., 2019*; *Sielaff et al., 2016*; *Wey-Fabrizius et al., 2014*). However, all members of the Acanthocephala are macroscopic, obligate endoparasites and highly differentiated in both morphological and molecular terms from other syndermatans. Finally, we included a selection of published protostome genomes as outgroups: three insects, a nematode, two tardigrades, five molluscs, two annelids, a brachiopod, platyhelminth, and an orthonectid (*Adams et al., 2000*; *Adema et al., 2017*; *Albertin et al., 2015*; *The C. elegans Sequencing Consortium, 1998*; *Gusev et al., 2014*; *Hashimoto et al., 2016*; *Luo et al., 2015*; *Mikhailov et al., 2016*; *Simakov et al., 2013*; *Yoshida et al., 2017*; *Young et al., 2012*; *Zhang et al., 2012*). These include the model species *Drosophila melanogaster* and *Caenorhabditis elegans*, high-quality genomes that are widely used to represent the other phyla, and two independent comparisons of desiccating and nondesiccating lineages (in tardigrades and *Polypedilum* midges), for evaluating the desiccation hypothesis. While we cannot sample all phyla in the same depth as the bdelloids, the protostome sample is unbiased by prior information on TE content and should broadly represent variation at the phylum level, sufficient to determine whether bdelloids do display unusual patterns of TE content.

## Abundant and diverse TEs in bdelloid genomes

To ascertain the repeat content of bdelloid genomes relative to other taxa in a consistent manner, we used the RepeatModeler and RepeatMasker pipelines to identify and classify repeats across all of the sampled genomes. The total proportion of the genome classified as repetitive ranged from ~19% to 45% across bdelloid genera, with variation within and between species (*Figure 2A*, *Figure 2—figure supplements 1* and *2*, *Figure 2—source data 1*, *Nowell et al., 2021*). Most of these are simple or unclassified repeats that do not belong to major TE superfamilies. While the precise nature of these unclassified repeats is not elucidated, an appreciable fraction (~7–27%, mean = 17%) are also annotated as protein-coding and thus may be derived from gene expansions or other duplications, while a further smaller fraction (<1%) are classified as class II miniature inverted-repeats (MITEs) (*Figure 2—source data 1*). For the bdelloid genomes sampled here, the proportion of the genome accounted for by known TEs (i.e. those classified into a known TE superfamily) ranged from 2.4% to 7.3% (mean = 4.9% ± 1.2 standard deviations [SD], median = 5.1%). Broken down by class and superfamily, the mean values are class I total = 2.09% ± 0.75 (PLEs = 0.59% ± 0.14; LTRs = 0.68% ± 0.26; and LINEs = 0.82% ± 0.47); class II total = 2.79 ± 0.8 (DNA transposons = 2.49% ± 0.77; rolling circles = 0.30 ± 0.11). These results are in broad agreement with previous estimates of TE content in bdelloids (*Arkhipova et al., 2017*; *Flot et al., 2013*; *Nowell et al., 2018*; *Simion et al., 2020*).

## No evidence for a major shift in TE composition in bdelloids

We first compared the bdelloid genomes to the other available syndermatan genomes. The sampled monogonont genomes are slightly more TE-rich than the bdelloids with a mean of 5.2% ± 1.5 SD

**Table 1.** Assembly statistics for 1 monogonont and 30 bdelloid rotifer reference assemblies presented in this study.

| Sample ID | Species name | SZ (Mb) | NN | N50 (kb) | L50 | AU (kb) | GC (%) | Gaps (kb) | Coverage (X) | Genome BUSCO score | CDS | Proteome BUSCO score | GenBank accession |
|---|---|---|---|---|---|---|---|---|---|---|---|---|---|
| Bc_PSC1 | *Brachionus calyciflorus* (Monogonont) | 116.7 | 14,869 | 18.5 | 1692 | 26.6 | 25.6 | 78 | 186 | C:96% [S:93%, D:3%],F:2% | 24,404 | C:98% [S:93%, D:5%],F:1% | GCA_905250105.1 |
| Ar_ARIC003 | *Adineta ricciae* | 135.6 | 4302 | 283.8 | 129 | 388 | 35.5 | 65 | 89 | C:97% [S:58%, D:39%],F:2% | 49,015 | C:97% [S:52%, D:45%],F:1% | GCA_905250025.1 |
| As_10x_p | *Adineta steineri* | 171.1 | 8257 | 200.1 | 163 | 394.5 | 29 | 206 | 198 | C:95% [S:67%, D:33%],F:2% | 50,321 | C:97% [S:58%, D:38%],F:2% | GCA_905250115.1 |
| As_ASTE804 | *Adineta steineri* | 160.3 | 9359 | 158.1 | 265 | 214.9 | 29.1 | 152 | 62 | C:95% [S:74%, D:22%],F:2% | 47,222 | C:98% [S:74%, D:24%],F:2% | GCA_905250045.1 |
| As_ASTE805 | *Adineta steineri* | 156.3 | 9008 | 169.6 | 245 | 226.4 | 29.2 | 129 | 65 | C:98% [S:77%, D:21%],F:1% | 43,986 | C:99% [S:72%, D:26%],F:1% | GCA_905250065.1 |
| As_ASTE806 | *Adineta steineri* | 160.3 | 7597 | 168.2 | 257 | 222.5 | 29.2 | 145 | 82 | C:96% [S:72%, D:24%],F:2% | 45,930 | C:98% [S:74%, D:24%],F:2% | GCA_905250035.1 |
| Dc_DCAR505 | *Didymodactylos carnosus* | 323.6 | 87,048 | 7.8 | 11,656 | 10.5 | 33.5 | 41 | 21 | C:86% [S:69%, D:17%],F:8% | 46,286 | C:88% [S:71%, D:18%],F:9% | GCA_905249995.1 |
| Dc_DCAR706 | *Didymodactylos carnosus* | 368.8 | 78,356 | 12 | 7695 | 19.1 | 33.5 | 13 | 76 | C:95% [S:70%, D:25%],F:2% | 46,863 | C:95% [S:71%, D:25%],F:2% | GCA_905250885.1 |
| Rd_10x_p | *Rotaria sordida* | 272.5 | 16,571 | 64.5 | 843 | 193.8 | 30.8 | 395 | 91 | C:94% [S:77%, D:19%],F:2% | 44,299 | C:95% [S:69%, D:26%],F:2% | GCA_905250005.1 |
| Rd_RSOR408 | *Rotaria sordida* | 252.9 | 20,315 | 57.6 | 1246 | 75.5 | 30.4 | 291 | 39 | C:94% [S:76%, D:19%],F:3% | 40,501 | C:97% [S:73%, D:24%],F:2% | GCA_905250875.1 |
| Rd_RSOR410 | *Rotaria sordida* | 252.6 | 19,518 | 60.9 | 1179 | 80 | 30.4 | 252 | 42 | C:95% [S:77%, D:18%],F:3% | 40,474 | C:98% [S:74%, D:24%],F:2% | GCA_905251635.1 |
| Rd_RSOR504 | *Rotaria sordida* | 251.3 | 22,067 | 53.1 | 1338 | 69.8 | 30.4 | 369 | 39 | C:94% [S:78%, D:16%],F:3% | 41,085 | C:96% [S:73%, D:23%],F:3% | GCA_905252715.1 |
| Rg_MAG1 | *Rotaria magnacalcarata* | 178.7 | 19,184 | 42 | 1077 | 62.4 | 32 | 402 | 58 | C:97% [S:81%, D:16%],F:1% | 40,318 | C:99% [S:76%, D:22%],F:1% | GCA_905261645.1 |
| Rg_MAG2 | *Rotaria magnacalcarata* | 181.1 | 22,216 | 39.7 | 1141 | 61 | 32 | 433 | 63 | C:98% [S:81%, D:17%],F:1% | 40,289 | C:99% [S:74%, D:26%],F:0% | GCA_905273325.1 |
| Rg_MAG3 | *Rotaria magnacalcarata* | 180.9 | 22,132 | 40.7 | 1142 | 60 | 32 | 508 | 60 | C:96% [S:80%, D:17%],F:1% | 40,740 | C:99% [S:77%, D:22%],F:0% | GCA_905319835.1 |
| Rg_RM15 | *Rotaria magnacalcarata* | 174 | 18,391 | 46.5 | 966 | 67.3 | 32 | 430 | 55 | C:96% [S:80%, D:16%],F:2% | 38,283 | C:99% [S:77%, D:22%],F:1% | GCA_905321285.1 |
| Rg_RM9 | *Rotaria magnacalcarata* | 173.8 | 19,520 | 44 | 999 | 64.7 | 31.9 | 594 | 51 | C:96% [S:80%, D:16%],F:1% | 38,404 | C:98% [S:76%, D:22%],F:1% | GCA_905321535.1 |
| Rp_RPSE411 | *Rotaria* sp. 'Silwood-2' | 296.5 | 30,050 | 102.5 | 381 | 691.1 | 31 | 247 | 35 | C:93%, [S:72%, D:21%],F:4% | 48,378 | C:95% [S:72%, D:23%],F:4% | GCA_905329745.1 |
| Rp_RPSE503 | *Rotaria* sp. 'Silwood-2' | 285.6 | 33,174 | 78.3 | 449 | 627.2 | 31.3 | 446 | 34 | C:91%, [S:75%, D:17%],F:5% | 48,269 | C:92% [S:72%, D:20%],F:7% | GCA_905330235.1 |

*Table 1 continued on next page*

Table 1 continued

| Sample ID | Species name | SZ (Mb) | NN | N50 (kb) | L50 | AU (kb) | GC (%) | Gaps (kb) | Coverage (X) | Genome BUSCO score | CDS | Proteome BUSCO score | GenBank accession |
|---|---|---|---|---|---|---|---|---|---|---|---|---|---|
| Rp_RPSE809 | *Rotaria* sp. 'Silwood-2' | 271.1 | 28,589 | 101.6 | 350 | 681.4 | 31 | 377 | 27 | C:93%, [S:76%, D:17%],F:4% | 47,010 | C:95% [S:74%, D:22%],F:4% | GCA_905330535.1 |
| Rp_RPSE812 | *Rotaria* sp. 'Silwood-2' | 264.1 | 34,498 | 80.8 | 403 | 616.2 | 31.1 | 428 | 27 | C:89%, [S:74%, D:15%],F:8% | 47,040 | C:90% [S:73%, D:17%],F:8% | GCA_905330805.1 |
| Rs_AK11 | *Rotaria socialis* | 149.2 | 6303 | 111.3 | 370 | 150.2 | 31.8 | 442 | 39 | C:97%, [S:80%, D:17%],F:1% | 34,844 | C:99% [S:75%, D:24%],F:1% | GCA_905331015.1 |
| Rs_AK15 | *Rotaria socialis* | 147.4 | 5030 | 134.7 | 305 | 177.6 | 31.8 | 423 | 37 | C:96%, [S:79%, D:18%],F:1% | 34,140 | C:98% [S:76%, D:23%],F:1% | GCA_905331295.1 |
| Rs_AK16 | *Rotaria socialis* | 147.4 | 4720 | 139.5 | 296 | 180.3 | 31.8 | 332 | 43 | C:97%, [S:80%, D:18%],F:0% | 33,717 | C:99% [S:76%, D:23%],F:1% | GCA_905331475.1 |
| Rs_AK27 | *Rotaria socialis* | 149.9 | 5952 | 123.7 | 343 | 159.8 | 31.8 | 458 | 36 | C:97%, [S:80%, D:17%],F:0% | 34,369 | C:99% [S:75%, D:24%],F:1% | GCA_905331485.1 |
| Rs_RS1 | *Rotaria socialis* | 151.1 | 6254 | 124.9 | 334 | 166.2 | 31.8 | 490 | 40 | C:97%, [S:80%, D:17%],F:0% | 33,937 | C:99% [S:77%, D:22%],F:1% | GCA_905331495.1 |
| Rw_10x_p | *Rotaria* sp. 'Silwood-1' | 310.4 | 16,995 | 211.8 | 211 | 126.2 | 31.1 | 534 | 53 | C:95%, [S:76%, D:20%],F:1% | 44,241 | C:97% [S:73%, D:24%],F:1% | GCA_905250055.1 |
| Rw_RSIL801 | *Rotaria* sp. 'Silwood-1' | 268.4 | 28,548 | 136.5 | 288 | 687.3 | 30.8 | 472 | 45 | C:94%, [S:77%, D:17%],F:4% | 41,574 | C:95% [S:75%, D:21%],F:5% | GCA_905331515.1 |
| Rw_RSIL802 | *Rotaria* sp. 'Silwood-1' | 249.9 | 21,286 | 153.4 | 238 | 702.3 | 30.7 | 451 | 42 | C:92%, [S:76%, D:16%],F:4% | 39,577 | C:94% [S:76%, D:18%],F:4% | GCA_905331505.1 |
| Rw_RSIL804 | *Rotaria* sp. 'Silwood-1' | 247.6 | 25,643 | 118.3 | 287 | 660.4 | 30.8 | 667 | 34 | C:94%, [S:78%, D:16%],F:3% | 41,139 | C:96% [S:78%, D:19%],F:3% | GCA_905331525.1 |
| Rw_RSIL806 | *Rotaria* sp. 'Silwood-1' | 294.1 | 29,968 | 132.4 | 333 | 681.9 | 30.8 | 500 | 31 | C:95%, [S:79%, D:16%],F:2% | 48,259 | C:97% [S:78%, D:19%],F:2% | GCA_905331535.1 |

Sequence statistics codes: SZ, total sequence length (Mb); NN, number of sequences; N50, N50 scaffold length (kb); L50, N50 index; AU, expected scaffold size (area under 'Nx' curve, kb). BUSCO score based on eukaryote set (n = 303); BUSCO codes: C, complete; S, complete and single copy; D, complete and duplicated; F, fragmented.

The online version of this article includes the following source data for Table 1:

**Source data 1.** Sample information and data counts.

**Source data 2.** Assembly statistics for 30 maximum-haplotig assemblies.

class I TEs and 2.5% ± 1.0 SD class II TEs (*Figure 2A*, *Figure 2—figure supplements 1* and *2*, *Figure 2—source data 1*, *Nowell et al., 2021*). Repeat content differs on average from the bdelloids mainly in the composition of two types of class I retrotransposons. First, the monogononts encode more LINE-like retroelements, which make up on average approximately 50% of their TE content compared with 16% in bdelloids. However, a high proportion of LINEs (~35% of total TE content) is also found in both isolates of *D. carnosus*, a deeply branching lineage sister to other bdelloid taxa included in the analyses. Second, monogononts encode fewer PLEs (~1%) than bdelloids (~12%). The most striking difference, however, relates to the genome of the acanthocephalan *P. laevis*, which encodes a substantially greater proportion of repeats than either the bdelloids or monogononts. In agreement with Mauer et al., we find ~66% of the *P. laevis* genome to be composed of repeats. The large majority are class I retrotransposons (~71% of the total TE content) from the LINE (~52%) and

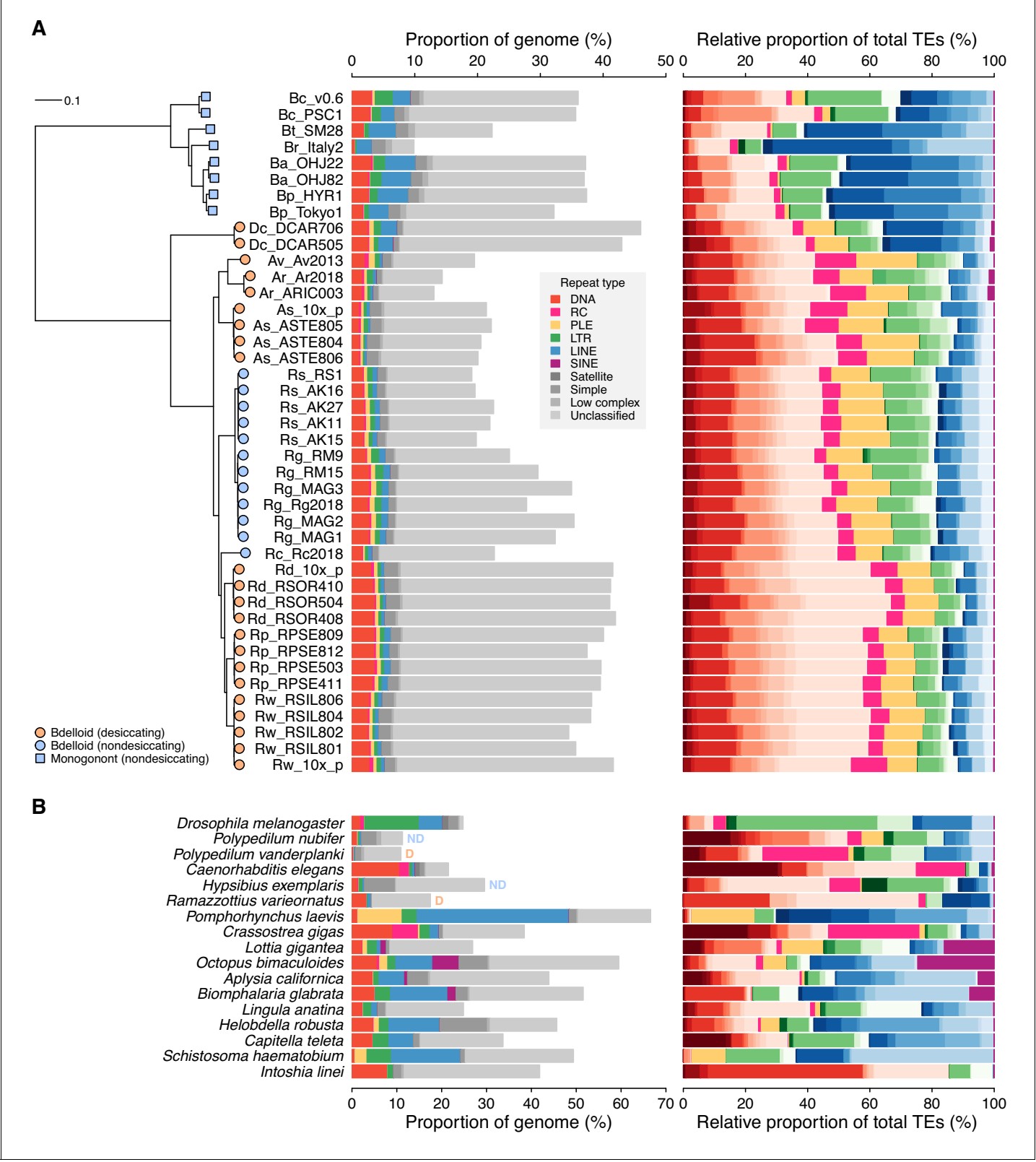

**Figure 2.** Repeat content and diversity in rotifer genomes. (**A**) Maximum likelihood phylogeny of eight monogonont (square symbols on tips) and 34 bdelloid (circles) genomes based on the concatenated alignment of a subset of core eukaryotic (BUSCO) proteins. Orange and blue tip colours indicate desiccating and nondesiccating taxa, respectively. Scale bar represents 0.1 amino acid substitutions per site. Species codes in tip names are: **Bc**, *Brachionus calyciflorus*; **Br**, *B. rotundiformis*; **Bt**, *B.* sp. 'Tiscar'; **Bp**, *B. plicatilis* HYR1; **Ba**, *B. asplanchnoidis*; **Dc**, *Didymodactylos carnosus*; **Av**, *Adineta*

*Figure 2 continued on next page*

*Figure 2 continued*

*vaga*; **Ar**, *A. ricciae*; **As**, *A. steineri*; **Rs**, *Rotaria socialis*; **Rg**, *R. magnacalcarata*; **Rc**, *R. macrura*; **Rd**, *R. sordida*; **Rw**, *R.* sp. 'Silwood-1'; **Rp**, *R.* sp. 'Silwood-2'. Repeat content is shown as the genome proportion (%) broken down by TE superfamily (middle panel), and relative proportion (%) of total known (i.e. classified) TEs (right panel), where colours represent TE superfamilies (see legend) and shades of colour represent different TE families within each superfamily. (B) Equivalent repeat content analysis in 17 protostome animal genomes, including the model species *D. melanogaster* and *C. elegans*, the recently published acanthocephalan rotifer *P. laevis*, and selected other species from across the protostome group. Two further examples of desiccating (orange 'D') and nondesiccating (blue 'ND') species pairs are shown: the insects *P. nubifer* and *P. vanderplanki* and the tardigrades *H. exemplaris* and *R. varieornatus*.

The online version of this article includes the following source data and figure supplement(s) for figure 2:

**Source data 1.** Repeat content data for rotifers and other animals.
**Source data 2.** Protein alignments and tree files for rotifer and protostome genomes.
**Source data 3.** Genome properties of 17 protostome animals included in the study.
**Source data 4.** Model output for phylogenetic models testing for significant shift in TE frequency on bdelloid stem branch compared to background, and MEDUSA-like test for significant shifts in rate of TE evolution on phylogenetic tree of bdelloids, monogononts, and outgroups.
**Source data 5.** Posterior mean and 95% credible intervals for the effect of desiccation ability on TE load from a phylogenetic linear model.
**Figure supplement 1.** Repeat content and diversity in 42 rotifer and 17 protostome animal genomes.
**Figure supplement 2.** TE content mapped as continuous trait onto rotifer phylogeny.
**Figure supplement 3.** TE content mapped as a continuous trait onto protostome phylogeny.
**Figure supplement 4.** Boxplots showing the raw data for repeat content in desiccating versus nondesiccating bdelloids, shown for (A) all bdelloids and (B) *Rotaria* lineages only.
**Figure supplement 5.** Phylogenetic signal (λ) in TE load variation among rotifer lineages, defined as the proportion of the total variance in TE load attributable to the phylogeny (*de Villemereuil and Nakagawa, 2014*).

PLE (~15%) superfamilies, and there are relatively few DNA transposons (~1.3%; *Figure 2B*). Thus, assuming that acanthocephalans are the closest relatives to bdelloids, the most parsimonious explanation for these broad scale differences in Syndermata is that the expansion of PLEs occurred in the ancestor to bdelloids and acanthocephalans, whereas the contraction of LINEs has occurred more recently, confined to a subset of sampled bdelloid genera.

The asexuality hypothesis predicts a significant decrease in TE content on the stem branch leading to the bdelloid clade, particularly for class I TEs (that include the LINEs). We tested this hypothesis formally by mapping the frequency of class I and II TEs onto a phylogenetic tree that included all our syndermatan genomes as well as the wider sample of protostomes as outgroups (*Figure 2—source datas 3* and *4*). Comparing a model with a single rate of evolution in TE frequency compared to an alternative model with a separate rate estimated on the stem branch of bdelloids, we found no evidence for a significant shift in either class I or class II TEs (log-likelihood ratio tests, chi-square = 1.4, 0.77 and p = 0.24, 0.38, respectively; *Figure 2—figure supplement 3* and *Figure 2—source data 4*). Next, fitting instead a model that searches for significant shifts in evolutionary rate across the tree (*Thomas and Freckleton, 2012*), without a prior hypothesis for where they should occur, the best-fit model recovered a major increase in class I TE frequency in the acanthocephalan branch (*Figure 2—figure supplement 3* and *Figure 2—source data 4*). A further shift was detected for class I TEs, but this was to a lower rate of evolution among bdelloid lineages (i.e. bdelloids are more uniform for class I TEs than expected compared to the background rate), rather than a shift to lower mean class I frequency across the clade as predicted by the hypothesis. In contrast, we did not detect any shifts along the major syndermatan branches for class II TEs (only two minor shifts detected: one within *A. steineri* and another in the oyster *Crassostrea gigas*; *Figure 2—figure supplement 3* and *Figure 2—source data 4*). Thus, it appears that bdelloid lineages have a lower rate of change in their overall class I TE content than expected compared to background rates of change among the other genomes.

As expected, total TE content varies widely across the protostome genomes from 0.8% in the insect *Polypedilum vanderplanki* to ~24% in the octopus and parasitic platyhelminth *Schistosoma haematobium* (*Figure 2B*). All the bdelloids we sampled encode relatively more TEs than both *Polypedilum* species but fewer than *Drosophila melanogaster*, *Caenorhabditis elegans*, annelid worms and some molluscs, and are intermediate with respect to other taxa. Thus, while this comparison is not a comprehensive analysis of TE content across all Protostomia, it is sufficient to show that bdelloids do not encode unusually fewer TEs than other animals. Also, while bdelloids do have lower frequencies of class I TEs (including LINEs) on average than either the monogononts or the

acanthocephalan we sampled, consistent with the earlier observation (*Arkhipova and Meselson, 2005*; *Arkhipova and Meselson, 2000*; *Dolgin and Charlesworth, 2006*; *Wright and Finnegan, 2001*), the numbers are still comparable to sexual organisms more broadly (e.g. *C. elegans*). Further-more, the bdelloid *D. carnosus*, which is a sister taxon to all other bdelloid species in this study and shares the same reproductive mode, did not show the same decrease in LINEs (~35%). Thus, we find no evidence that bdelloid rotifers have an unusual pattern of TE composition compared to other ani-mals and conclude that the simple expectations of TE evolution under the hypothesis of long-term asexuality (i.e. either runaway proliferation or complete elimination) are not met.

## No evidence for lower TE loads in desiccating bdelloids

The desiccation hypothesis posits that TE numbers may be kept in check via the action of DSB-repair processes during recovery from desiccation. Our study includes 11 nondesiccating bdelloid samples encompassing three obligately aquatic species (*R. macrura*, *R. magnacalcarata* and *R. socialis*), while the remaining samples were isolated from ephemeral ponds or moss and must undergo frequent cycles of desiccation and rehydration to survive. Contrary to the prediction that TE load should be reduced in desiccating species, there is little overall difference in TE proportions between desiccat-ing and nondesiccating lineages (mean = 4.8% ± 1.3 *vs.* 5.0% ± 0.9 respectively). Broken down by TE superfamily, desiccating taxa have relatively more DNA transposons, simple, low complexity, and unclassified repeats, and relatively fewer PLE, LTR, and LINE retroelements, with the biggest differ-ences seen between *Rotaria* lineages (*Figure 2—figure supplement 4*). However, based on two independent shifts in desiccation ability within our sample (see phylogeny in *Figure 2A* and *Fig-ure 2—source data 2*), results from a Bayesian mixed-effects modelling approach that controlled for phylogenetic relationships showed no significant correlations between desiccation ability and TE load, for either overall proportion or for any individual TE superfamily (p>0.05 in all cases; *Figure 2—source data 5*). For most TE superfamilies, the strength of the phylogenetic signal (λ) was close to 1 (*Figure 2—figure supplement 5*), consistent with a high fit of the data to the phy-logeny under a Brownian motion model as would be expected if TE load evolves neutrally along branches of the phylogeny (*Pagel, 1994*; *Szitenberg et al., 2016*). Thus, large-scale differences in TE content between lineages appear to be consistent with the action of genetic drift, except for the observed decrease in the rate of evolution of class I TEs among lineages of bdelloids reported above.

Adding two further comparisons of desiccating versus nondesiccating species within our wider sample of animals, to increase the power of our test, still yielded no evidence for a consistent effect (p>0.05 after correction for multiple testing in all cases; *Figure 2—source data 5*). In chironomid midges, the desiccation-tolerant *P. vanderplanki* encodes substantially fewer TEs than its nondesic-cating sister species *P. nubifer*, as predicted (0.8% and 2.2% respectively, although this rises to ~11% in both species when all repeats are included). In tardigrades, however, the desiccation tol-erant *Ramazzottius varieornatus* encodes a greater proportion of TEs than *Hypsibius exemplaris* (4.3% and 2.8%, respectively), which does not survive desiccation without extensive conditioning (*Hashimoto et al., 2016*), although the trend is reversed when all repeats are included due to a large fraction of simple repeats in *H. exemplaris*. We therefore find no consistent evidence for the hypothesised link between anhydrobiosis and reduced TE load in bdelloids or beyond.

## TE transposition is recent and ongoing

Having found no evidence for either the asexuality or desiccation hypothesis, we explored a range of possible reasons why the predictions from simple theory are not met. One possibility is that TEs are present in bdelloid genomes but do not replicate autonomously or are otherwise inactivated or 'fossilised' within the host genome. To investigate this, we first generated divergence 'landscapes' for identified TE copies within each genome, using the de novo RepeatMasker results. TE landscapes measure the amount of sequence divergence between each TE copy and a consensus derived from all copies in its family (*Smit et al., 2013*). Histograms of the resulting Kimura distances (*K*-values; *Kimura, 1980*) provide insights into the evolutionary history of TE activity (*Chalopin et al., 2015*; *Kapusta and Suh, 2017*; *Shao et al., 2019*).

TE landscapes for the three diploid (10x Genomics) assemblies of *A. steineri*, *R.* sp. 'Silwood-1' and *R. sordida* show that TE divergence is bimodal but strongly zero-inflated (*Figure 3*). A large

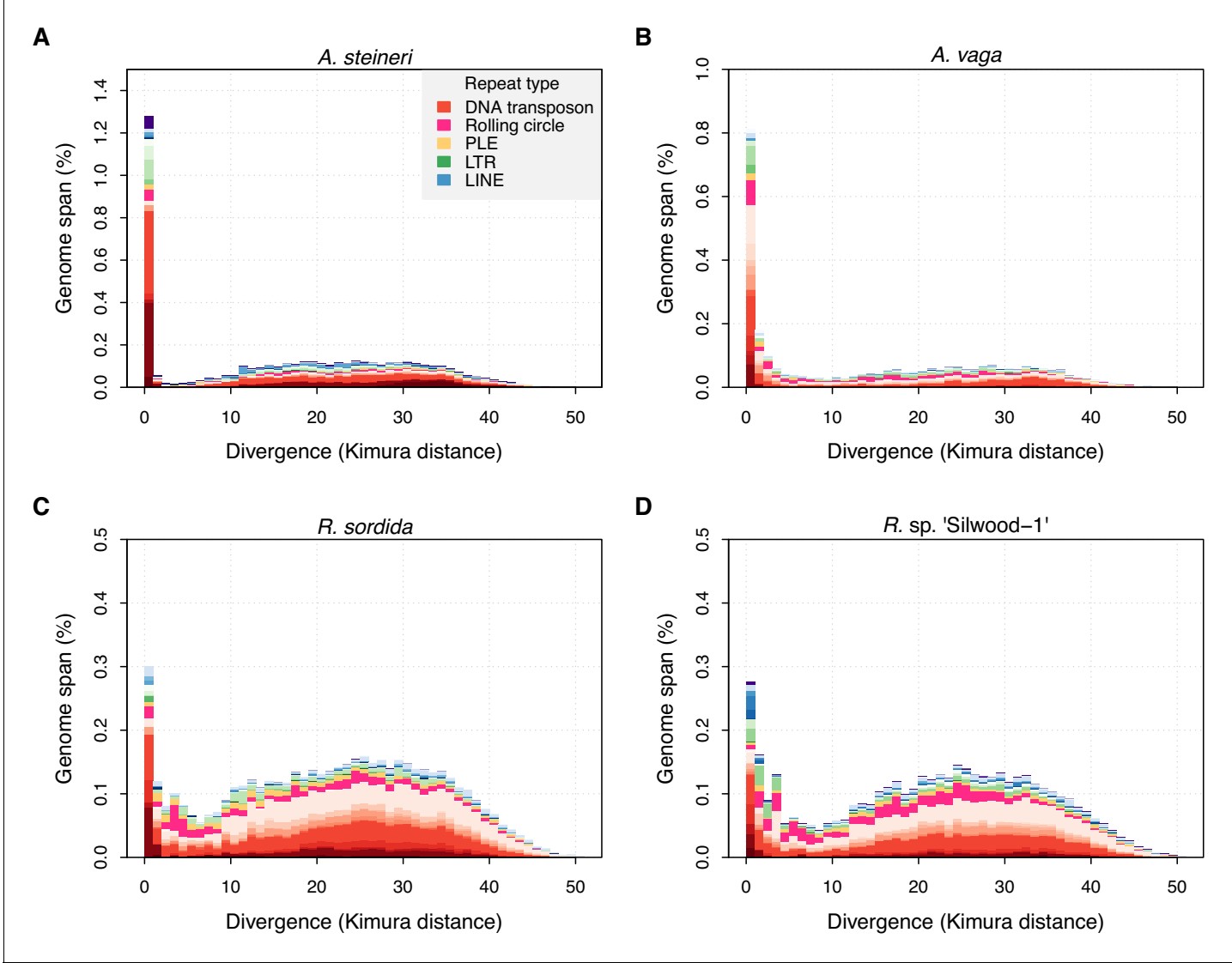

**Figure 3.** TE divergence landscapes for selected genomes. The *X*-axes show the level of divergence (Kimura substitution level, CpG adjusted) between each identified TE copy and the consensus sequence for that TE family (the inferred ancestral copy). Thus, if newly arising TE copies evolve neutrally, the amount of divergence is a proxy for the time since its duplication, with older copies accumulating more substitutions and appearing further to the right. The *Y*-axis shows the proportion of the genome occupied by each bin. Colours represent TE superfamilies (see legend) and shades of colour represent different TE families within each superfamily. Data are shown for the 10x Genomics diploid assemblies of *A. steineri*, *R.* sp. 'Silwood-1' and *R. sordida* compared to the published genome of *A. vaga*. Note different scales on some *Y*-axes.

The online version of this article includes the following source data and figure supplement(s) for figure 3:

**Source data 1.** Mapping results for species with available RNA-seq data, showing read coverage for annotated TEs based on RNA-seq data mapped to the corresponding genome.

**Figure supplement 1.** TE divergence landscapes for individual *Brachionus* genomes.

**Figure supplement 2.** TE divergence landscapes for individual *Adineta* genomes.

**Figure supplement 3.** TE divergence landscapes for individual *D. carnosus* and *R. sordida* genomes.

**Figure supplement 4.** TE divergence landscapes for individual *R. macrura* and *R. magnacalcarata* genomes.

**Figure supplement 5.** TE divergence landscapes for individual *R. socialis* genomes.

**Figure supplement 6.** TE divergence landscapes for individual *R.* sp. 'Silwood-1' and *R.* sp. 'Silwood-2' genomes.

**Figure supplement 7.** TE divergence landscapes for selected species constructed using REPET (unfiltered, with default parameters).

number of TE copies have very low or no divergence from the consensus ($K$-value $\leq$1%). Assuming a molecular clock for nucleotide substitutions within duplicated TEs, such elements represent recent duplications that are highly similar to their progenitor copy, consistent with recent transposition of an active element. In proportion, most of these belong to class II DNA transposon superfamilies (in red), but the spike of zero divergence is also present for class I retrotransposons (in blue and green). An older, broader mode is seen around a $K$-value of 20–30% that probably reflects historical TE transpositions and/or a signal from the tetraploid genome structure present in all bdelloids sequenced to date. The same pattern was observed in the haplotype-resolved assemblies of *A. vaga* (*Flot et al., 2013*) and *A. ricciae* (*Nowell et al., 2018*), and was generally present but less pronounced in the other 'maxhap' assemblies depending on the repeat pipeline applied (*Figure 3—figure supplements 1–7*). Further support for ongoing TE activity is also found in transcriptomic data, available for a handful of species (*A. ricciae*, *A. vaga*, *R. magnacalcarata*, *R. socialis*, and *R. sordida*), which shows evidence of transcription for approximately one third of annotated TEs on average, depending on the species and the TE superfamily (*Figure 3—source data 1*).

To evaluate recent TE activity further, we took advantage of within-population sampling of multiple individuals for a subset of our bdelloid species. We developed a simple method to identify insertion sites for class I LTR retrotransposons (LTR-Rs) and assess their presence or absence polymorphism within populations. We chose LTR-Rs because the majority of LTR families insert randomly in the host genome (*Burt and Trivers, 2009*; *Eickbush and Malik, 2002*), meaning that the neighbouring genome sequence provides a unique marker for a given insertion event without the problem of homoplastic or 'recurrent' insertions caused by insertion–site specificity (*Belshaw et al., 2004*). We constructed a library of such insertion markers ('LTR-tags') for all full-length LTR-Rs (i.e. those with long-terminal repeats present at both the 5′ and 3′ ends of the element) detected in our genomes, and then searched for their presence or absence in the other samples. For a given LTR-tag identified in genome *A*, the presence of a contiguous alignment in genome *B* indicates that the same insertion is shared between *A* and *B*.

For a set of 161 high-confidence and non-redundant LTR-Rs identified in the single-individual samples, alignment contiguity for each LTR-tag versus each of the other genomes was scored using a read-mapping approach (see Materials and methods), resulting in a pairwise matrix of presence/absence scores (*Figure 4A*, *Figure 4—figure supplement 1* and *Figure 4—source data 1*). High scores for LTR insertion-site presence correlated strongly with the phylogeny, resulting in an average score of ~0.9 within species compared to <0.1 between species and a clear visual signal along the diagonal of *Figure 4A*. Very few LTR insertion sites were shared between bdelloid species. While some absences could reflect loss rather than gain, the restriction of nearly all LTR insertion sites to single species indicates that they have been gained during the separate evolutionary history of that species.

LTR-R insertions also vary between individuals within the same species, indicating recent transposition events and the potential for ongoing fitness consequences for the host. One case-study is illustrated for *R. magnacalcarata* (*Figure 4B*, *Figure 4—figure supplement 1*). The individuals RM9 and RM15 share an LTR-R insertion that is not present in conspecifics. Aligning the regions of the genome assemblies containing these LTR-tags indicates that an 8.1 kb LTR-R has inserted into a protein-coding sequence in the lineage leading to RM9 and RM15. It has introduced a premature stop codon to a gene that encodes a protein (7479 residues) of unknown function but with partial similarity to midasin, an ATPase essential to ribosome biosynthesis in several model eukaryotes (*Garbarino and Gibbons, 2002*; *Li et al., 2019*). In RM9 and RM15, the predicted product is substantially truncated (to 6025 residues) by the element insertion. Despite the potential fitness consequences, RM9 and RM15 have evidently persisted for some time since, because they differ at approximately 0.5% of single-nucleotide sites across the 8.1 kb LTR element itself. A possible explanation is that both the RM9 and RM15 assemblies also contain a scaffold with an empty insertion site, which we interpret as an intact version of the coding sequence spanned by the LTR insertion (represented in *Figure 4B* by the partial matches on scaffolds RM9 16719 and RM15_07127, respectively). If the insertion is hemizygous, an uninterrupted homologous copy of the affected gene might mask or reduce the effect of the mutation.

Thus, these data contradict the idea that bdelloid TEs are inactive. All TE superfamilies show a substantial fraction of copies at low-divergence, indicative of recent proliferation, while direct evidence of TE transcription is observed in those species with available RNA-seq data. Moreover, there

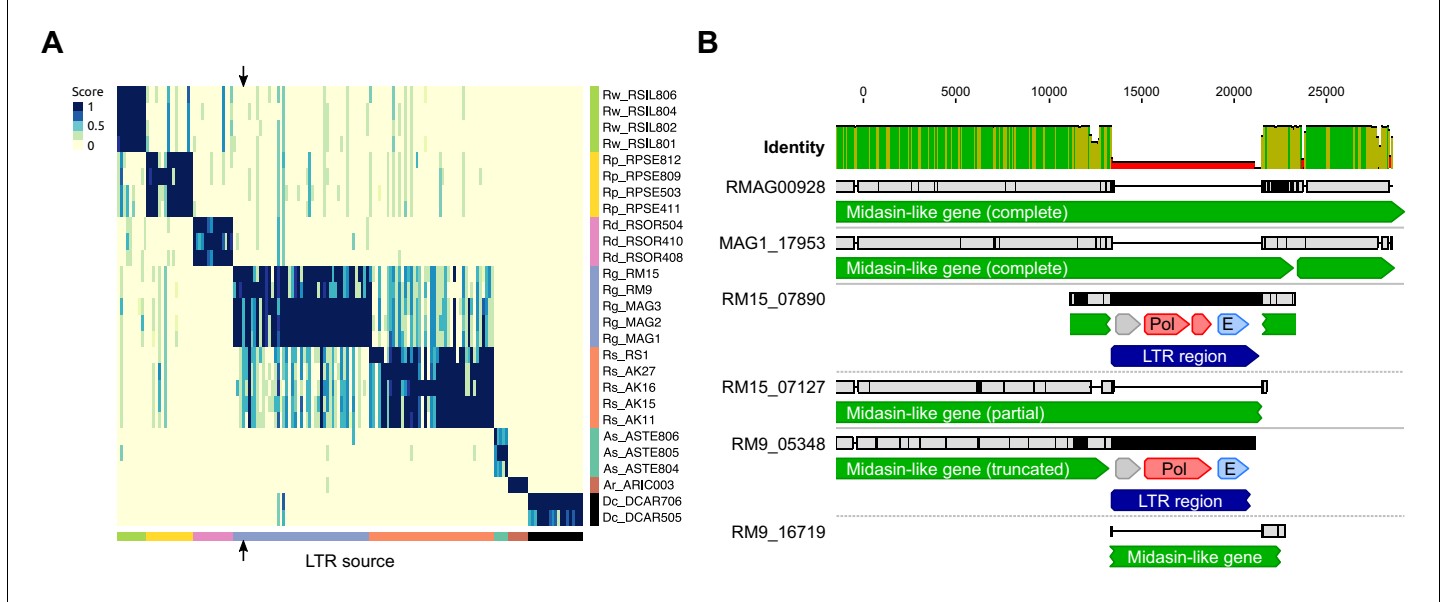

**Figure 4.** LTR insertion-site polymorphism in bdelloid species. (**A**) Columns represent 161 LTR-Rs identified across bdelloid samples, arranged by genome of origin (see colours at bottom and side). Support for the presence of a given LTR-R at a specific insertion site in each genome is scored from 0 (absent, yellow) to 1 (present, dark blue), where a score <0.5 is strong evidence for absence (see Materials and methods for details). Arrows demark the column corresponding to the LTR-R example shown in (B). (**B**) Nucleotide alignment of region around an LTR-R insertion (blue) identified in RM9 (scaffold 05348) and RM15 (scaffold 07890), alongside their putative homologous scaffolds (scaffolds 16719 and 07127 respectively) that do not show the insertion. Scaffolds from Rg2018 (RMAG00928) and MAG1 are also shown for comparison. Predicted CDS with similarity to Pol and Env proteins are shown in red and light blue. The LTR-R is most likely a member of the *TelKA* family, based on sequence similarity.

The online version of this article includes the following source data and figure supplement(s) for figure 4:

**Source data 1.** LTR-tag fasta file and mapping data.
**Source data 2.** Analysis of recombination in LTR-tag presence/absence data.
**Figure supplement 1.** LTR-R polymorphism details.
**Figure supplement 2.** The median and 95% Highest Posterior Density interval of the frequency of sexual recombination affecting the presence/absence of LTR polymorphisms in *R. magnacalcarata* and *R. socialis*.

are multiple cases of insertion-site polymorphism within species, and at least one case where a recent retroelement insertion into a predicted protein-coding sequence seems likely to have potential fitness consequences.

## No evidence that cryptic recombination helps to limit the spread of LTR-Rs

A second possible explanation for the apparent discrepancy between bdelloid TE content and theory is that bdelloids in fact possess cryptic inter-individual recombination, either through undetected sex or some alternative form of gene transfer. We therefore tested for a signature of recombination among polymorphic LTR-R insertion sites within species. Under strict clonality, the pattern of presence and absence across LTR-R loci should be nested and compatible with only mutational gain and loss at each site. In contrast, in a sexual, outcrossing population, variation should be shuffled among loci. LTR-Rs provide a powerful test of these predictions because random insertion makes independent origins of the same LTR-R insertion site highly unlikely.

In every species with multiple samples, we found that variation in polymorphic TEs is perfectly nested, with a consistency index in parsimony reconstruction of 1. Furthermore, in the two species with multiple parsimony-informative characters, *R. socialis* and *R. magnacalcarata*, we found a significantly positive index of association of presence and absences among LTR-R insertion sites, as expected with clonal inheritance (*Figure 4—figure supplement 2* and *Figure 4—source data 2*). Approximate Bayesian Computation with simulations of expected patterns under varying frequencies of sexual reproduction showed that strictly clonal evolution could not be rejected. While this

test uses a restricted set of markers, and so should not be viewed as a test of recombination for the whole genome or species, it does support clonal inheritance of LTR-R loci and finds no evidence that inter-individual recombination helps to limit or facilitate their spread. Nevertheless, local LTR-LTR recombination within genomes, leading to solo LTR formation, may act to bring the copy number down (*Flot et al., 2013*), and certain LTR elements, particularly those encoding *env*-like proteins (*Rodriguez et al., 2017*), may still move horizontally between hosts independent of any host DNA exchange.

## Bdelloids experience similar selective constraints on TEs as do other species

Another possibility is that the selective environment for TEs is different in bdelloids than in other animals, thereby shifting their TE profiles compared to simple theory. For instance, bdelloids might tolerate insertions within genes unusually well, owing to redundancy arising from tetraploidy or multiple gene copies (*Eyres et al., 2012*; *Hur et al., 2009*; *Mark Welch et al., 2008*). First, we explored the genomic 'environment' of TE insertions and their potential effects on genome function. Differences in the location of TE insertions might reveal differential costs and benefits compared to other taxa. We first compiled a high-confidence list of class I retrotransposons by searching for proteins with significant similarity to the reverse transcriptase (RT) domain found in all retrotransposons. Phylogenies of the resulting alignments showed a diverse array of RTs in all species, most of them full-length (in terms of conserved subdomain presence) and clustered within the three primary retrotransposon superfamilies—PLEs, LTRs, and LINEs (*Figure 5A*, *Figure 5—figure supplement 1* and *Figure 5—source data 1*). Many (but not all) clustered within families previously identified in *A. vaga*. The elevated LINE content in *D. carnosus* in comparison to other bdelloids is mostly due to high numbers of elements in the Soliton clade and to the presence of CR1-Zenon and Tad/I/Outcast clades, the latter characterised by the RNase H domain.

We then surveyed genome features surrounding these TEs. In 50 kb windows surrounding each class I TE identified above, we counted the occurrence and span of three features of interest: other (non-TE) genes, other (non-focal) TEs, and the telomeric repeat 'TGTGGG' (identified from *A. vaga* and supported in other rotifers; *Gladyshev and Arkhipova, 2007*; *Figure 5—source data 2*). Phylogenetic linear models showed that, relative to a set of core metazoan (BUSCO) genes, the regions surrounding PLE, LINE, and LTR TEs all showed significant decreases in gene density, but significant increases in the density of both other TEs and telomeric repeats (p<0.001 in all cases, *Figure 5B–D* and *Figure 5—source data 3*). However, we found no significant differences in the density of these three genomic features surrounding retroelements in monogononts versus bdelloids or desiccating versus nondesiccating bdelloid species (p>0.05 in all cases; *Figure 5—source data 3*). These results are consistent with previous findings that TEs are concentrated in subtelomeric regions of rotifer genomes (*Gladyshev and Arkhipova, 2010a*), a bias that is presumably due to selection against insertions at or near functioning genes, but do not suggest any major differences based on either asexuality or desiccation ability. Thus, it appears that most TE insertions are costly in bdelloid rotifers, as in other taxa, and that selection leads to their concentration outside of gene-rich regions.

As a second source of selective constraints, we tested for evidence of selection against ectopic recombination (ER). ER is argued to be a major cost of TEs in sexual taxa, but its effects derive from chromosomal abnormalities during meiosis, which should be lacking in bdelloids. Because the rate of ER increases with both the number of elements and their length (*Montgomery et al., 1987*), the strength of purifying selection is expected to be strongest against longer TEs at higher copy number (*Bourgeois and Boissinot, 2019*; *Petrov et al., 2003*). Work in vertebrates has shown that selection remains even for truncated TEs that are non-functional, implicating the ER model over other possible fitness costs, such as direct deleterious effects of TE-derived RNAs or proteins (*Song and Boissinot, 2007*; *Xue et al., 2018*). Thus, two testable predictions arise: first, that bdelloids should have longer TEs than sexual taxa (under the hypothesis that ER is absent in bdelloids because of a lack of meiosis), and second, that nondesiccating bdelloids should have longer TEs than desiccating bdelloids (under the hypothesis that ER may still occur when chromosomes pair during the repair of DSBs). Phylogenetic linear models comparing TE length distributions found no significant difference between monogononts and bdelloids, or between desiccating and nondesiccating bdelloids (p>0.05 in all cases, *Figure 6A*, *Figure 6—source datas 1* and *2*). Thus, while the precise estimation of TE lengths will no doubt improve with increasing assembly contiguity, the current data provide no

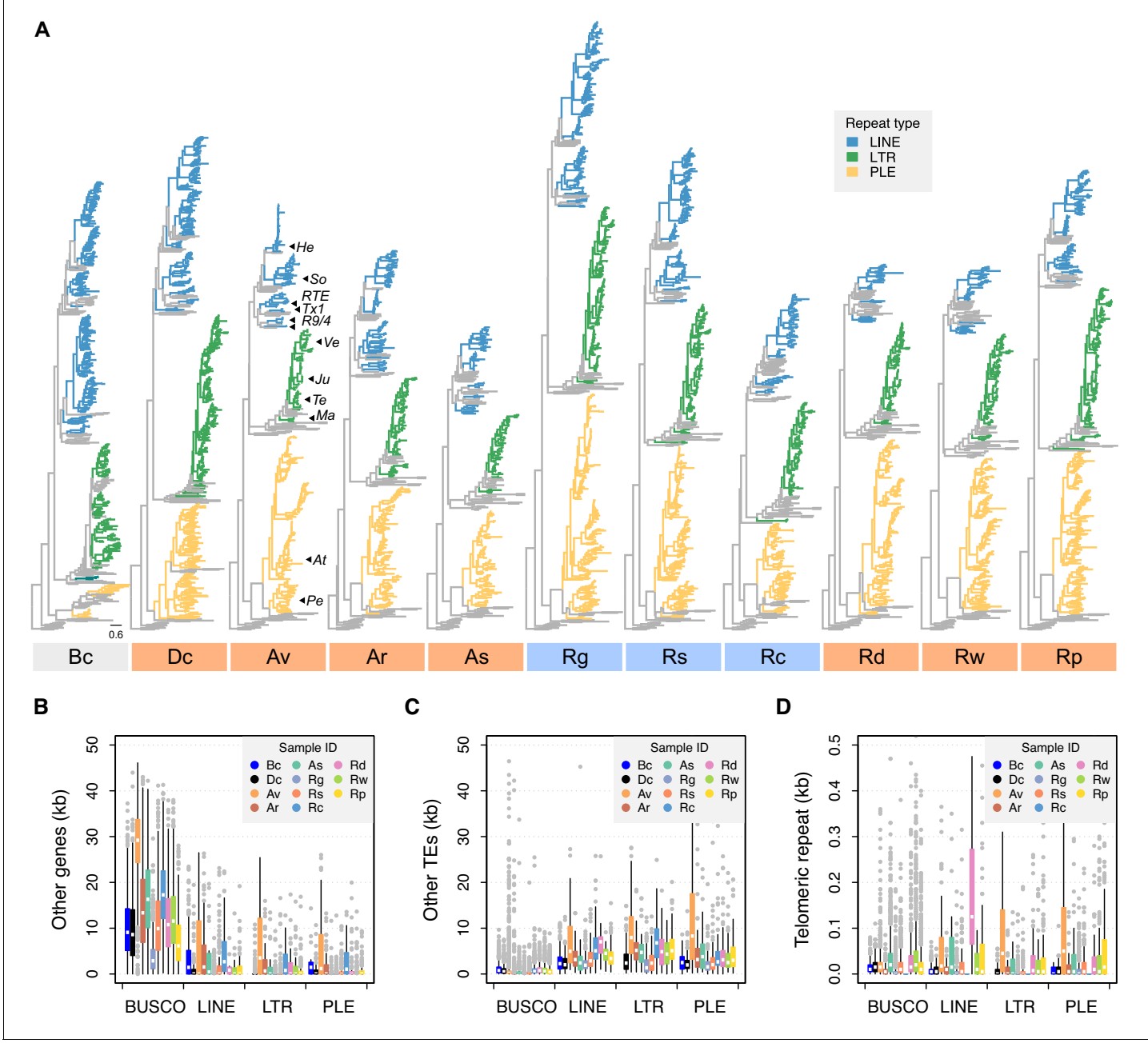

**Figure 5.** Phylogenetic diversity and genomic context of reverse transcriptase (RT) genes. (**A**) For each phylogeny, coloured branches represent identified rotifer-encoded RT copies and grey branches represent the core RT sequences from which the hidden Markov model (HMM) was built (see *Figure 5—figure supplement 1* for core RT tree details). Colours indicate the major superfamilies. Previously characterized retrotransposons are indicated on the *A. vaga* tree (He, *Hebe*; So, *Soliton*; RTE, *RTE*; Tx1, *Tx1*; R9/4, *R9* and *R4*; Ve, *Vesta*; Ju, *Juno*; Te, *TelKA*; Ma, *Mag*; At, *Athena*; Pe, *Penelope*). All phylogenies are rooted on the branch separating the bacterial retrons. Scale bar represents 0.6 amino acid substitutions per site. Desiccating and nondesiccating species are indicated with orange and blue, as previously. Species codes: Bc, *B. calyciflorus* PSC1; Dc, *D. carnosus* DCAR706, Av, *A. vaga* Av2013; Ar, *A. ricciae* ARIC003; As, *A. steineri* ASTE805; Rg, *R. magnacalcarata* MAG3; Rs, *R. socialis* AK11; Rc, *R. macrura* Rc2018; Rd, *R. sordida* RSOR408; Rw, *R.* sp. 'Silwood-1' RSIL806; Rp, *R.* sp. 'Silwood-2' RPSE503. The genomic context in which RT genes reside is then described based on proximity to three other features: (**B**) other genes (that do not overlap with any TE annotation), (**C**) other TEs, and (**D**) telomeric repeats ('TGTGGG'; that do not overlap with any coding region) as identified in *A. vaga*. For each plot, a 50 kb window is drawn around the focal TE and the total span (kb) of each feature within the window is counted, broken down per sample ID (coloured boxes, see legend) per TE superfamily (X-axis groups). Boxplots show the median (band), interquartile range (box) and minimum/maximum values (whiskers; outliers are shown in grey). The equivalent data for BUSCO genes (metazoan set) are also shown for comparison. The same set of individuals are shown in (**B–D**) as for (**A**). Average values (mean ± SD) across monogononts and bdelloids (desiccating and nondesiccating) are provided in *Figure 5—source data 4*.

*Figure 5 continued on next page*

*Figure 5 continued*

The online version of this article includes the following source data and figure supplement(s) for figure 5:

**Source data 1.** Reverse-transcriptase alignments and phylogenies.

**Source data 2.** Identification of putative telomeric repeats in rotifer genomes.

**Source data 3.** Posterior mean and 95% credible intervals for the effects of asexuality and desiccation ability on the density of three features (other genes, other TEs and telomeric repeats) surrounding LINE, LTR, and PLE class I TEs, compared to BUSCO genes.

**Source data 4.** Mean and standard deviation (SD) for the span (kb) of features 'other genes', 'other TEs' and 'telomeric repeats' occurring in 50 kb windows around genes of type BUSCO, LINE, LTR, or PLE, averaged across monogononts versus bdelloids and desiccating versus nondesiccating bdelloids.

**Figure supplement 1.** Maximum likelihood phylogeny for diverse reverse transcriptase domains from across the tree of life.

evidence of changes in TE length linked to asexuality (when compared to monogononts) or to desiccation ability within bdelloids.

A final prediction of selection against ER is that there should be a negative correlation between TE frequency and length, as is observed in *Drosophila* (*Petrov et al., 2003*) and humans (*Song and Boissinot, 2007*). For both monogononts and bdelloids, the majority of identified TEs are short (<1 kb), presumably due to partial matches or degraded copies. Nonetheless, we observe a sharp decline in copy number as mean TE length increases above ~0.5 kb, and a distinct lack of longer elements at higher copy numbers (*Figure 6B*). In vertebrates, previous work has suggested a lower

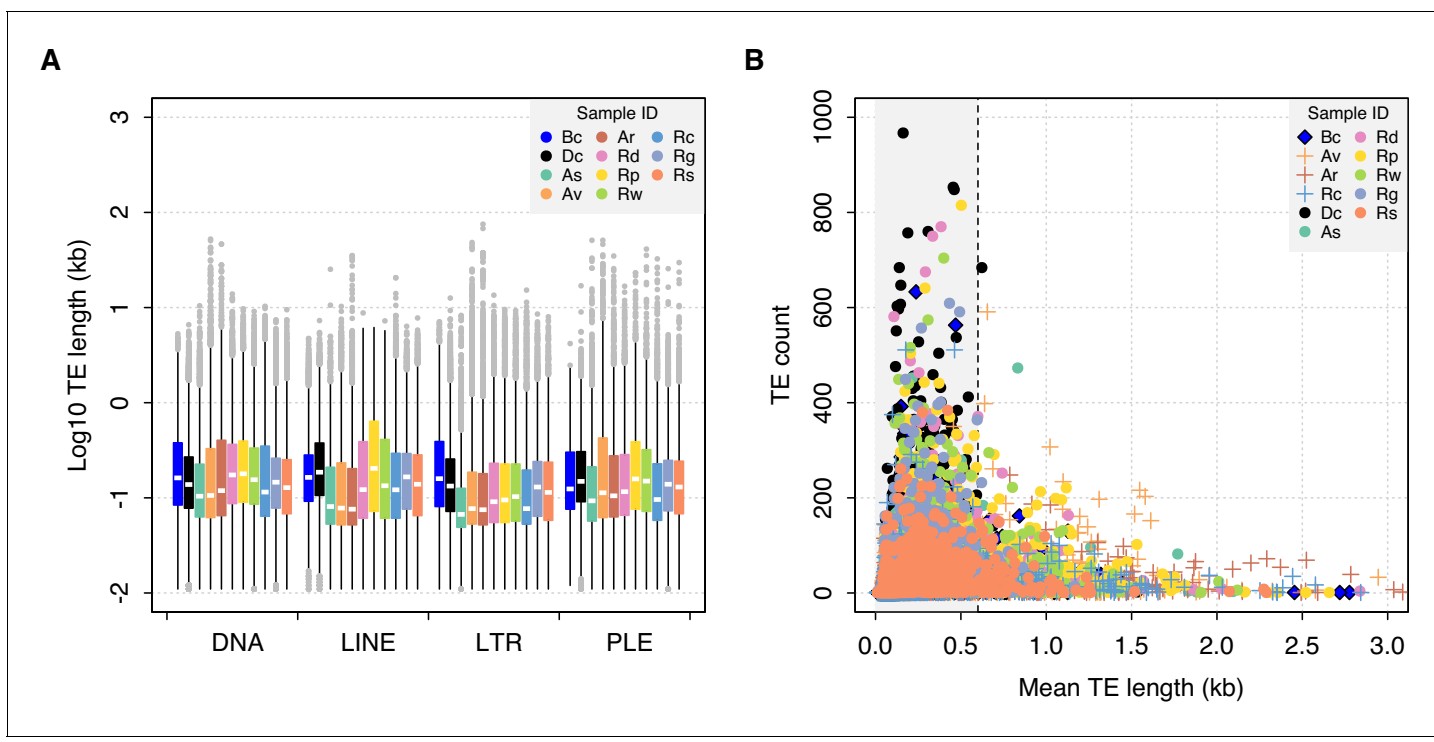

**Figure 6.** TE length dynamics. (**A**) Distribution of TE length for selected syndermatan samples decomposed into the major TE superfamilies (DNA transposons, LINE, LTR, and PLE retrotransposons). Boxplots show the median (band), interquartile range (box) and minimum/maximum values (whiskers; outliers are shown in grey). Species codes: **Bc**, *B. calyciflorus* PSC1 (monogonont); **Dc**, *D. carnosus* DCAR706; **As**, *A. steineri* ASTE804; **Av**, *A. vaga* Av2013; **Ar**, *A. ricciae* Ar2018; **Rd**, *R. sordida* RSOR408; **Rp**, *R.* sp. 'Silwood-2' RPSE411; **Rw**, *R.* sp. 'Silwood-1' RSIL801 (desiccating bdelloids); **Rc**, *R. macrura* Rc2018; **Rg**, *R. magnacalcarata* MAG1; **Rs**, *R. socialis* AK11 (nondesiccating bdelloids). An equivalent plot including the acanthocephalan *P. laevis* is shown in *Figure 6—figure supplement 1*. (**B**) Relationship between mean TE length per TE family (*X*-axis) and copy number (i.e. the number of TEs identified within each family; *Y*-axis). The same set of individuals are shown as for (A). A dashed line is drawn at 0.6 kb, given as the length threshold below which the rate of homologous ectopic recombination is negligible in mice.

The online version of this article includes the following source data and figure supplement(s) for figure 6:

**Source data 1.** TE length raw data.

**Source data 2.** Posterior mean and 95% credible intervals for the effects of asexuality desiccation ability on TE length.

**Figure supplement 1.** Distribution of TE lengths including the acanthocephalan *P. laevis*.

threshold of ~0.6–1 kb under which ectopic recombination does not operate (*Cooper et al., 1998*; *Song and Boissinot, 2007*). Thus, the observed patterns in rotifers are consistent with the hypothesis that longer elements above a certain length threshold are selected against more strongly due to the deleterious effects of ectopic recombination. Nonetheless, the pattern is the same in both desiccating and nondesiccating bdelloid representatives as well as the monogonont *B. calyciflorus* and the acanthocephalan *P. laevis* (*Figure 6—figure supplement 1*), suggesting that selection against longer TEs at higher copy number is a general feature in Syndermata, regardless of reproductive mode or desiccation ability.

## Expansion and diversification of TE silencing pathways in bdelloids

The final possible explanation that we consider for why bdelloid TE profiles do not match with simple theory is that molecular pathways that defend against TEs might be unusually expanded in bdelloids. We explored this possibility by characterizing copy number variation for three well-known gene families with direct roles in TE suppression via RNA interference (RNAi): (1) Argonaute proteins of both the Ago and Piwi subfamilies, the core effectors of RNAi gene-silencing that form complexes with various classes of small RNA (*Höck and Meister, 2008*; *Juliano et al., 2011*); (2) Dicer, an RNase III–family protein that cleaves double-stranded RNA (dsRNA) molecules from 'target' genes into shorter fragments that are subsequently incorporated into Argonaute complexes (*de Jong et al., 2009*; *Ghildiyal and Zamore, 2009*); and (3) RNA-dependent RNA polymerase (RdRP), an RNA replicase that synthesises secondary small interfering RNAs (siRNAs) that amplify the silencing response (*Ghildiyal and Zamore, 2009*; *Zong et al., 2009*).

Based on hidden Markov model (HMM) matches of key domains to the predicted proteomes of the Illumina 'haploid' assemblies (in which homologous copies are largely collapsed but homoeologous copies are both present), we detected an average of 21.5 putative Argonaute, 3.9 Dicer and 38.9 RdRP copies in bdelloid genomes (*Figure 7A*, *Figure 7—source data 1*). For comparison, in monogonont genomes we found 7.5, 3.5 and 2.5 average copies for Argonaute, Dicer, and RdRP, respectively, while in a selection of eukaryotic species (see Materials and methods) the average copy number for these genes was 5.8, 1.9, and 0.6. Thus, it appears that bdelloid genomes contain a substantially larger number of both Argonaute and RdRP (but not Dicer) genes, relative to either monogononts or eukaryotes more generally. Phylogenies of identified copies of the focal genes themselves revealed a number of divergent clades, particularly for Argonaute and RdRP (*Figure 7B–D*, *Figure 7—figure supplement 1* and *Figure 7—source datas 1* and *2*). Additional analysis of RdRP using a much larger phylogeny of eukaryote RdRP genes (*Pinzón et al., 2019*) showed that the majority of bdelloid copies (the clades designated 'RDR I' and 'RDR II' in *Figure 7B*) do not cluster within the major metazoan RdRP clades, and are seemingly quite divergent from any known RdRP in the sample (*Figure 7—figure supplement 2*). Thus, the most likely evolutionary scenario to explain the diversity of RdRP copies in bdelloids is that they represent the retention of an ancestral lineage (or perhaps an ancient HGT gain) that has undergone subsequent duplication and divergence within the bdelloid lineage.

We used the birth model of gene duplication (*Hahn et al., 2005*; *Han et al., 2013*) to explicitly test for significant expansions in family size on our tree of Syndermata and representative outgroup protostome genomes. We found a highly significant shift in copy number on the stem branch leading to the bdelloid clade, reconstructed as a gain of 9 Argonaute copies and 27 RdRP copies (*Figure 8A*, *Figure 8—source data 1*). At the taxonomic level of class and above, the model also detects a significant increase in Argonaute copies in the branch leading to the nematode *C. elegans*, driven by the well-known 'worm-specific' Ago (WAGO) genes found in nematodes (*Buck and Blaxter, 2013*; *Shi et al., 2013*; *Yigit et al., 2006*). To check that these inferences were not affected by variation in ploidy among genomes, we plotted the copy number of functional domains in both bdelloid and monogonont proteomes relative to the set of reference eukaryotes, across all predicted proteins. In monogononts, we find no evidence for increased numbers of RNAi genes, with domain counts for PAZ, PIWI (the key domains of Argonaute proteins), Dicer, and RdRP being distributed close to the 1-to-1 line across all proteins (*Figure 8B*, *Figure 8—source data 2*). In bdelloids, however, there is a general shift to a higher gene copy number across all protein domains due to tetraploidy (*Figure 8C*). To test whether the number of RNAi genes are outliers relative to this shifted distribution, we applied a conservative correction factor of 0.5 to bdelloid abundance scores, to account for uncollapsed homoeologous copies, and calculated the position of each gene family on a

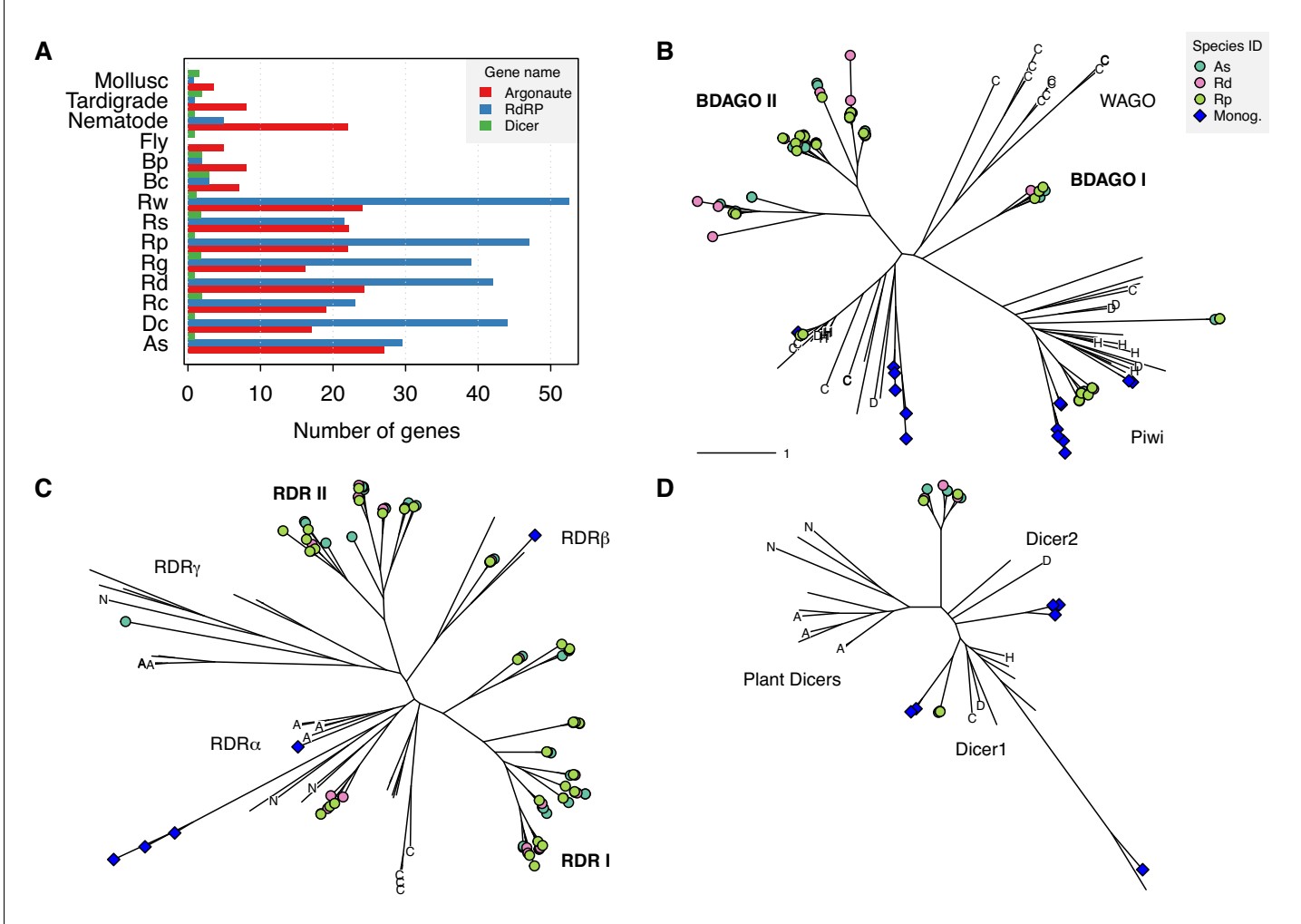

**Figure 7.** Expansion of TE silencing pathways in bdelloid rotifers. (A) Copy number variation for RNAi gene families Argonaute (Ago/Piwi, red), RNA-dependent RNA polymerase (RdRP, blue) and Dicer (green) in bdelloids compared to other protostome groups. Proteins are identified based on the presence of key identifying domains (see Materials and methods). Species codes for rotifers: **Bc**, *B. calyciflorus*; **Bp**, *B. plicatilis* HYR1; **Dc**, *D. carnosus*; **As**, *A. steineri*; **Rg**, *R. magnacalcarata*; **Rs**, *R. socialis*; **Rc**, *R. macrura*; **Rd**, *R. sordida*; **Rw**, *R.* sp. 'Silwood-1'; **Rp**, *R.* sp. 'Silwood-2'. Maximum likelihood unrooted phylogenies are then shown for (B) Argonaute, (C) RdRP and (D) Dicer gene copies identified in *A. steineri*, *R. sordida* and *R.* sp. 'Silwood-1' 10x haploid assemblies, aligned with orthologs from representative species from across the eukaryotes. Blue symbols indicate copies identified in the monogonont *B. plicatilis*, and letters on tips show selected reference species to aid visual orientation: 'C', *C. elegans*; 'H', human; 'D', *D. melanogaster*; 'N' *N. crassa*; 'A', *A. thaliana*. Some clade names are also shown where relevant; 'WAGO' indicates the worm-specific cluster of Ago genes in the Argonaute phylogeny. 'BDAGO I/II' and 'RDR I/II' indicate putative bdelloid-specific clades of Argonaute and RdRP proteins, respectively.

The online version of this article includes the following source data and figure supplement(s) for figure 7:

**Source data 1.** RNAi gene family expansions.
**Source data 2.** Argonaute, Dicer, and RdRP alignments and phylogenies.
**Figure supplement 1.** Argonaute, RdRP, and Dicer phylogeny details.
**Figure supplement 2.** Extended RdRP phylogeny.

distribution of scores normalised by the average abundance in eukaryotes (see Materials and methods). Thus, an abundance ratio >0 indicates a greater number of copies, on average, in bdelloids relative to eukaryotes. The resulting abundance ratios ($n$ = 4894) are distributed approximately normally with mean = 0.03 ± 0.88 SD, with values for Dicer = 0.26, PAZ = 0.46, PIWI = 0.87 and RdRP = 2.3, corresponding to the 66th, 75th, 87th, and 98th percentiles, respectively (*Figure 8D*). Thus, even accounting for tetraploidy in bdelloids, the RdRP domain appears to be particularly expanded relative to other eukaryotes, while the PAZ and PIWI domains of Argonaute proteins are also overrepresented in bdelloids. This pattern is not found in the monogononts *B. calyciflorus* or *B.*

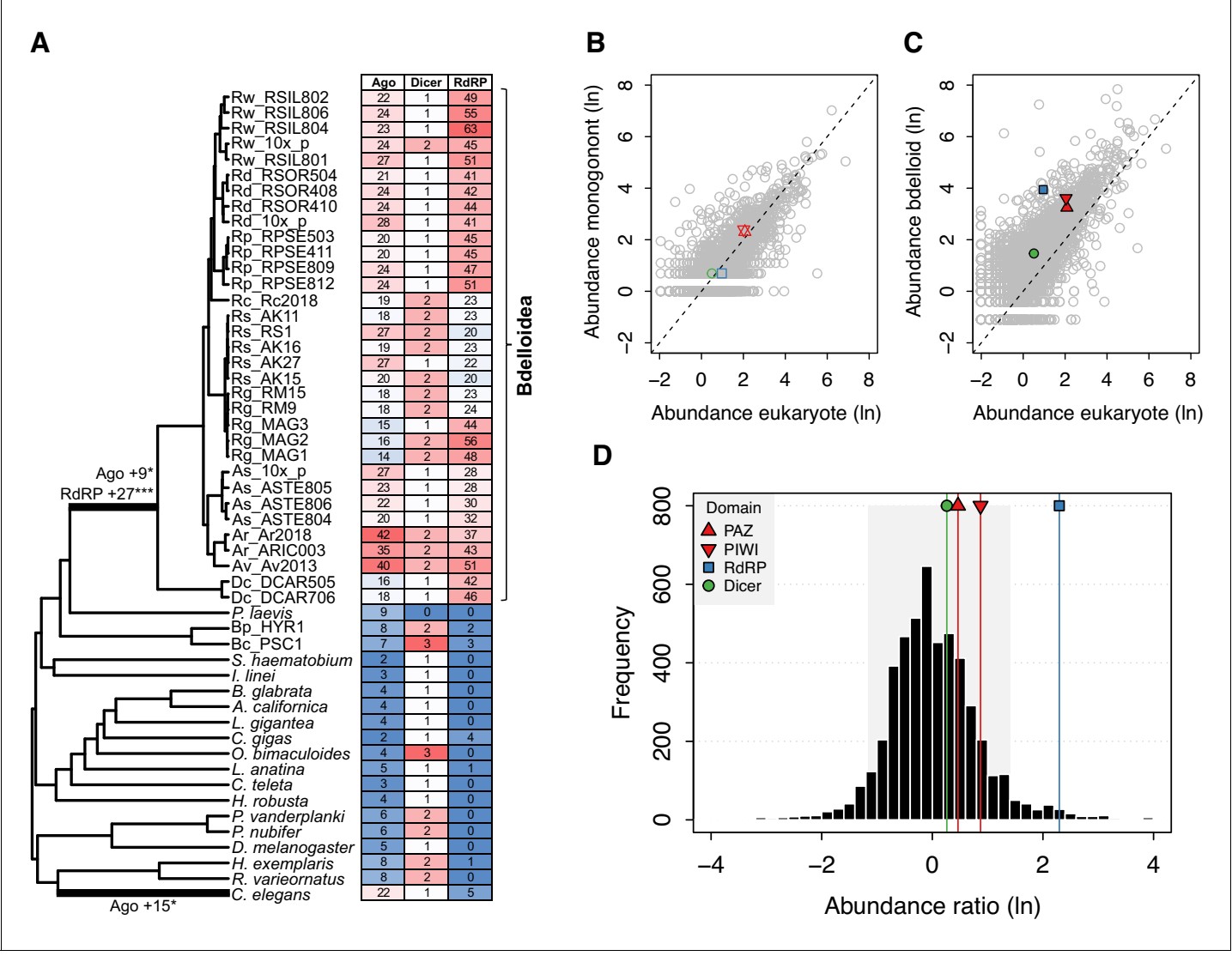

**Figure 8.** Evidence for significant expansion of Argonaute and RdRP gene families in bdelloids. (**A**) Evidence for significant expansion of Argonaute (+9 copies; p<0.05) and RdRP (+27 copies; p<0.001) on the stem branch leading to the bdelloid clade. A significant expansion of Argonaute genes (+15 copies; p<0.05) is also found on the branch leading to the nematode *C. elegans*, corresponding to the nematode WAGO genes. Phylogeny is based on the concatenated alignment of a subset of core eukaryotic (BUSCO) proteins for this set of 59 taxa (*Figure 2—source datas 3* and *4*). Numbers of Ago, Dicer, and RdRP proteins (based on HMM hits of key domains to predicted proteomes) are shown in the table, shaded by their relative abundance. The relatively lower numbers seen for *R. socialis* and some *R. magnacalcarata* individuals is probably an artefact of assembly 'collapse' due to low heterozygosity in these genomes. (**B**) Comparative protein-domain abundance plot. Each point represents a Pfam domain ID, with (log$_e$) average abundance (i.e. count) in the reference eukaryote set shown on the *X*-axis and (log$_e$) abundance in the monogonont *B. plicatilis* on the *Y*-axis. The positions of the PAZ and PIWI (key domains of Argonaute; red up/down triangles), RdRP (blue square), and Dicer (green circle) domains are highlighted. Dashed line indicates the 1-to-1 relationship. (**C**) Equivalent plot for bdelloids, where the *Y*-axis shows the (log$_e$) average abundance for the *A. steineri*, *R. sordida*, and *R.* sp. 'Silwood-1' 10x haploid assemblies. Note that the average abundance for all Pfam entries is shifted above the 1-to-1 line due to the ancient genome duplication in all bdelloids, such that many genes are found in double copy (i.e. homoeologs) even in a 'haploid' representation. (**D**) Comparative protein-domain abundance plot for bdelloids versus eukaryotes (see Materials and methods). Entries to the right of the mean of the distribution are overrepresented in bdelloids with respect to eukaryotes. The shaded area represents the 5% and 95% quantiles of the distribution, and the scores for the PAZ, PIWI, Dicer, and RdRP domains are indicated (see legend).

The online version of this article includes the following source data for figure 8:

**Source data 1.** CAFE model fitting birth rate model to gene family evolution of RNAi pathways.
**Source data 2.** Pfam count data.

*plicatilis* HYR1, nor is there evidence for it in the (unannotated) acanthocephalan genome (*Figure 7— source data 1*), suggesting that, while our comparison is limited to a relatively small set of reference eukaryotes, the diverse repertoire of Argonaute and RdRP genes appears to be a unique feature of bdelloids.

## Conclusions

We show that bdelloids encode a rich diversity of TEs from both class I (retroelements) and class II (DNA transposons), many of which show evidence of recent activity within populations. We do not find evidence of major shifts in TE content on the stem branch leading to bdelloids. These findings reject the idea that bdelloids are deficient or unusual in their TE content or diversity and are at odds with the predictions of population genetic theory for TEs in long-term asexuals. One possible resolution is that theory is missing some component or assumption. It is possible that parameter space exists that permits intermediate levels of TEs in an asexual population, perhaps sustained by high rates of horizontal transfer even among elements without adaptations for autonomous horizontal travel (e.g. LINEs). This would be consistent with genomic evidence for extensive horizontal capture of non-metazoan DNA by bdelloids. Alternatively, some TEs might have been co-opted to provide beneficial functions, which is hypothesised to explain the unusually large and complex *Terminon* repeats (*Arkhipova et al., 2017*; *Arkhipova and Yushenova, 2019*). The role of *Terminons* might be clarified by future investigation of acanthocephalan genomes, which seem to share the stem PLE expansion. Other TEs may have evolved strong site-specificity to neutral genome regions to mitigate negative effects of transposition. This idea is supported by the preference shown for insertions into gene-poor regions that are probably at or near the telomeres, although it seems unlikely that the full complement of bdelloid TEs have accumulated in this way.

An alternative resolution is that the assumption of no recombination is not met in bdelloids. The desiccation hypothesis, for example, proposed that intragenomic recombination during the repair of DSBs caused by desiccation could provide a mechanism to keep TE numbers in check. We found no evidence here that overall TE loads or activity were lower in desiccation-tolerant bdelloid rotifers versus nondesiccating species, nor in two further comparisons of desiccating and nondesiccating tardigrade and insect species. It is possible that the overall effect of desiccation on TEs might be dual: while repair of a DSB within a TE via non-homologous end-joining would likely result in its inactivation (thus acting to reduce TE load), an efficient repair system would enhance repair of DSBs that arise during transposition of cut-and-paste DNA TEs that leave a DSB behind upon excision (thus allowing an increased TE load). It is also possible that TE proliferation is kept in check in nondesiccating species by mechanisms such as mitotic recombination. Alternatively, the assumption of strict clonality could be incorrect and there could be some hidden mechanism of sex or inter-individual recombination that facilitates TE removal. We found no evidence for its action on TE polymorphism here, but further clarification of the genetic system in bdelloids will be needed to resolve this question.

Overall, we find that the evolution of TE load in rotifers is largely consistent with genetic drift. Our analyses did find two significant shifts in bdelloid rotifers related to TE evolution and possible defence mechanisms, however. First, we found evidence for lower rates of evolution in class I TEs content among bdelloid lineages than observed among the other lineages we sampled. While increased depth of sequencing of the other lineages is needed to confirm this result, especially across Syndermata, lower rates of change might indicate stronger control on class I TEs in bdelloids than in the other animals. Second, we detected significant expansion of certain RNAi gene silencing pathways in bdelloids, with RdRP genes especially being present in greater number and diversity than described in any other animal to date (*Pinzón et al., 2019*; *Zong et al., 2009*). While the precise origins and functions of these divergent Ago and RdRP clades are yet to be elucidated, we speculate that such an extended arsenal of TE defence genes might offer enhanced protection against the deleterious effects of TE activity, particularly if bdelloid populations cannot keep TEs in check through sexual processes. It has been shown in *A. vaga* that piwi-interacting small RNAs (piRNAs) target both TEs and putatively 'foreign' genes (i.e. non-metazoan genes gained via HGT) (*Rodriguez and Arkhipova, 2016*), which are unusually frequent in bdelloid genomes. Thus, one possibility is that bdelloids require an extensive RNAi system to defend against invasion from horizontally transferred TEs, or indeed other transferred genes, particularly if the level of exposure or rate of import is higher than in other animals (*Flot et al., 2013*). For example, multiple copies of

RdRP may be required for the amplification of secondary piRNAs, since *A. vaga* apparently lacks the canonical Piwi-mediated mechanism of piRNA generation (known as the 'ping-pong' cycle) (*Rodriguez and Arkhipova, 2016*). In addition, there might be alternative functions for these pathways, such as defence against infectious viruses as reported in plants (*Guo et al., 2019*; *Xie et al., 2001*), or the recognition of 'self' versus 'non-self' RNA and multigenerational (i.e. inherited) epigenetic memory as reported in the nematode *C. elegans* (*Buck and Blaxter, 2013*; *Gilbert, 2017*; *Shirayama et al., 2012*). Future work is required to elucidate the functional significance of these expansions.

## Materials and methods

### Rotifer sampling and culture

For most samples, individual rotifers were collected from permanent and temporary freshwater habitats around Imperial College London's Silwood Park campus (Ascot, UK), between May 2015 and February 2019. Three samples (*R. magnacalcarata* RM9 and RM15, and *R. socialis* RS1) were collected from a freshwater spring in Fontaneto d'Agogna, Italy in 2016. Animals belonging to *R. sp.* 'Silwood 1' and *R. sp.* 'Silwood 2' were isolated from a temporary pond where we had previously sampled the desiccation-tolerant species *Rotaria tardigrada* Ehrenberg 1832 (*Eyres et al., 2015*). These two undescribed species (abbreviated respectively as 'Rw' and 'Rp') closely resemble *R. tardigrada* in morphology, but phylogenomic (*Figure 2A*) and marker-based analyses (data not shown) clearly delineate them as two molecular entities distinct from each other and from any publicly available sequence assigned to *R. tardigrada*. They are taken to be desiccation-tolerant because this habitat patch regularly dries out, and at least one living individual was observed in a rehydrated sample of mud that had been completely dried in the laboratory and stored at 40% relative humidity for 34 days. Although we focused on the genera *Adineta* and *Rotaria*, we also included two individuals from the desiccation-tolerant species *Didymodactylos carnosus*. Preliminary phylogenetic data had identified this as a distant outgroup to the focal genera, useful in rooting phylogenetic trees and as a further independent datapoint to test the generality of conclusions about bdelloids.

A total of 26 samples were submitted for single-individual, whole genome sequencing; for these, DNA was extracted using either a Chelex preparation (Bio-Rad InstaGene Matrix) or a QIAamp DNA Micro Kit (Qiagen), and whole-genome amplified using a REPLI-g Single Cell kit (Qiagen) before sequencing on either Illumina NextSeq500 at the Department of Biochemistry, University of Cambridge (Cambridge, UK), or Illumina HiSeq X at Edinburgh Genomics, University of Edinburgh (Edinburgh, UK). For *A. ricciae* ARIC003, DNA was extracted from ~200 animals descended from a single individual before whole-genome amplification. For *B. calyciflorus* PSC1, individuals for DNA extractions were derived from an individual isolate from a laboratory stock population previously isolated from field-collected resting eggs (*Becks and Agrawal, 2011*). DNA was extracted from ~5000 starved individuals using a phenol-chloroform protocol and sequenced on the Illumina NextSeq500 at the Max Planck Institute for Evolutionary Biology. Three 10x Genomics Chromium 'linked reads' libraries were generated for *A. steineri*, *Rotaria* sp. 'Silwood-1' and *R. sordida*; for these, high-molecular-weight DNA was extracted from thousands of animals reared clonally from a single wild-caught animal, without whole-genome amplification, using the Chromium Demonstrated Protocol 'HMW gDNA Extraction from Single Insects' (https://support.10xgenomics.com/permalink/7HBJe-Zucc80CwkMAmA4oQ2). Linked-read libraries were constructed at the Centre for Genomics Research, Liverpool, UK, before sequencing on the HiSeq X at Edinburgh Genomics. Further details on rotifer sampling, DNA extraction and sequencing are provided in *Table 1—source data 1*.

### Biological replicates

To check the repeatability of the whole-genome amplification (WGA), sequencing, assembly and analysis pipelines, we included several samples that were either biological replicates of the same rotifer clone, or where high-quality genomes were available for the same clone from unamplified source material. Specifically, for *Rotaria* sp. 'Silwood-2' we isolated two consecutive offspring from the same wild-caught mother and conducted WGA, sequencing, assembly and analysis for these sisters independently (as Rp_RPSE411 and Rp_RPSE503). From the same clonal laboratory line of *Rotaria* sp. 'Silwood-1' that was used for 10x Genomics DNA preparation, we isolated two more

individuals and processed each independently using the WGA workflow (as Rw_RSIL801 and Rw_RSIL802). Finally, we applied the WGA method to DNA from *A. ricciae*, for which a previous assembly was available from unamplified DNA (*Nowell et al., 2018*) on the same clonal culture and included this replicate in downstream analyses alongside the earlier reference assembly.

## Data filtering and genome assembly

We generated two assembly versions for each of the single-individual rotifer samples. The 'reference' assemblies were scaffolded and polished to result in haploid assemblies with improved contiguity. The 'maximum haplotig' ('maxhap') assemblies instead retained highly similar contigs that might otherwise be removed during assembly polishing. Our pipeline is outlined as follows.

For the Illumina libraries, raw sequence data were filtered for low quality bases and adapter sequence using BBTools v38.73 'bbduk' (*Bushnell, 2014*), and error corrected using BBTools 'tadpole'. Data quality was inspected manually using FastQC v0.11.5 (*Andrews, 2015*) aided by MultiQC (*Ewels et al., 2016*) visualisation. For the *A. steineri*, *R.* sp. 'Silwood-1' and *R. sordida* linked-read libraries, data were assembled into haploid ('pseudohap') and diploid ('megabubbles') genome representations using the 10x Genomics proprietary software Supernova v2.1.1 (*Weisenfeld et al., 2017*) and further scaffolded with ARKS v1.0.4 (*Coombe et al., 2018*). All raw sequencing data have been deposited in the relevant International Nucleotide Sequence Database Collaboration (INSDC) databases under the Study ID PRJEB43248 (see *Table 1—source data 1* for run accessions and counts for raw and filtered data).

For the single-individual samples, an initial assembly was generated using SPAdes v3.13.0 (*Bankevich et al., 2012*) with default settings. Contaminating reads from non–target organisms, identified based on aberrant GC content, read coverage and/or taxonomic annotation, were then identified and removed using BlobTools v1.1.1 (*Buchfink et al., 2015*; *Laetsch and Blaxter, 2017*). For *R. magnacalcarata* and *R. socialis* samples, resultant haplotigs were then collapsed using Redundans (*Pryszcz and Gabaldón, 2016*) with default settings before scaffolding and gap filling with SSPACE v3.0 and GapCloser v1.12, respectively (*Boetzer et al., 2011*; *Luo et al., 2012*). For *A. steineri*, *R.* sp. 'Silwood-1' and *R. sordida* single-individual samples, the scaffolding step was performed with RaGOO v1.1 (*Alonge et al., 2019*; *Li, 2018*), using the matching 10x Genomics haploid ('pseudohap') assembly as a reference (contigs from *R.* sp. 'Silwood-2' were scaffolded using the *R.* sp. 'Silwood-1' 10x assembly), specifying the '-C' parameter to prevent concatenation of unaligned contigs. Scaffolded assemblies were subjected to further rounds of BlobTools to remove any additional sequences derived from non-target organisms. These assemblies were designated the reference set described above.

For the maxhap assemblies, filtered fastq files were first generated by mapping the original (trimmed and error-corrected) sequencing reads to each reference genome, using the 'outm=filtered_R#.fq' functionality of BBTools 'bbmap', and then reassembled with SPAdes, increasing the final kmer value to 121. Assembly metrics were summarised using 'calN50.js' (*Li, 2020*), which reports the 'expected scaffold size' (AU) as an alternative metric of assembly contiguity that is less biased than N50 (defined as the area under the cumulative genome span versus contig length graph, equivalent to the expected scaffold size for a randomly chosen assembly location; *Salzberg et al., 2012*). Gene-completeness scores for core eukaryotic ($n = 303$) and metazoan ($n = 978$) genes were calculated for all assemblies using BUSCO v3.0.2 (*Simão et al., 2015*) with default settings. Reference and maxhap assemblies for *B. calyciflorus* PSC1 are the same, due to a lack of appropriate data for scaffolding.

## Gene prediction

Gene prediction was performed on reference assemblies using one of three approaches, depending on the availability of RNA-seq data. For *B. calyciflorus*, *A. ricciae*, and all *R. magnacalcarata*, *R. socialis*, and *R. sordida* assemblies, published RNA-seq data (*Boschetti et al., 2012*; *Eyres et al., 2015*; *Hanson et al., 2013*) were downloaded from NCBI Sequence Read Archive (SRA), quality-trimmed using BBTools 'bbduk' with default settings and aligned to the genomic scaffolds using STAR v2.7.3a (*Dobin et al., 2013*) with the option 'twoPassMode Basic'. Aligned BAM files were then provided to BRAKER v2.1.2 (*Barnett et al., 2011*; *Hoff et al., 2016*; *Hoff et al., 2019*; *Stanke et al., 2008*; *Stanke et al., 2006*) with default settings for gene prediction. For *A. steineri*, *R.*

sp. 'Silwood-1' and *R.* sp. 'Silwood-2' assemblies, RNA-seq data from a related species (*A. ricciae* and *R. magnacalcarata* respectively) were used instead, aligned using BBTools 'bbmap' with the options 'maxindel=200k minid=0.5', before gene prediction with BRAKER as above. Finally, for the distantly related *D. carnosus*, BRAKER was run using gene-model parameters estimated from BUSCO analysis of the genomic scaffolds. The quality of predicted proteins was assessed using BUSCO in protein mode. Intragenomic divergence between homologous gene copies and collinearity was calculated as for *Nowell et al., 2018*. Genome assemblies and gene predictions were converted to EMBL format using EMBLmyGFF3 v2 (*Norling et al., 2018*), and have been deposited at DDBJ/ENA/GenBank under the Study ID PRJEB43248 (see *Table 1* and *Table 1—source data 2* for individual GenBank accessions).

## Rotifer phylogeny

Evolutionary relationships among new genomes and published genomes of rotifers were determined using a core-genome phylogenomics approach based on the BUSCO eukaryotic gene set. For genomes from species with very high intragenomic homologous divergence (*A. ricciae* and *A. vaga*), redundancy among multiple BUSCO gene copies was removed by selecting the copy with the highest BUSCO score, using the script 'BUSCO_collapse_multicopy.pl' (*Nowell, 2020*). One-to-one co-orthologs found in at least 95% of the samples were then identified using the script 'BUSCO_phylo-genomics.py' (*McGowan et al., 2020*). Protein sequences were aligned using Clustalo (*Sievers et al., 2011*) and concatenated in Geneious Prime v2020.1.2 (*Kearse et al., 2012*). The full alignment was checked by eye and sections with ambiguous alignment within the bdelloid clade were removed across all sequences to avoid aligning potential paralogs or homoeologs. Translation errors arising from annotation issues in specific bdelloid genomes were identified by obvious mismatches to the consensus of closely related genomes, and the affected residues were deleted in the affected genome only. Potential alignment issues within the monogonont clade were less obvious owing to the substantial genetic divergence from bdelloids and the smaller number of genomes and replicates, so sections of ambiguous alignment within the monogonont clade were removed across all monogononts to avoid arbitrary decisions about the relative reliability of different genomes. The final alignment length was 44,675 residues in length, with ~66% average pairwise identity. A maximum-likelihood phylogeny was then estimated using IQ-TREE v1.6.12 with automatic model selection (*Nguyen et al., 2015*; *Kalyaanamoorthy et al., 2017*). The best fit model according to Bayesian Information Criterion (BIC) was a revised JTT matrix (*Kosiol and Goldman, 2005*) with amino acid frequencies estimated from the data, allowing for a proportion of invariable sites and four gamma-distributed rate categories (JTTDCMut+F+I+G4). Branching support was assessed using SH-aLRT and ultrafast bootstrap sampling ('-alrt 1000 -bb 5000') (*Guindon et al., 2010*; *Hoang et al., 2018*). A similar approach was used to generate a phylogeny for the full sample of 59 taxa (rotifers + protostomes), used for phylogenetic modelling, but without manual correction for alignment errors or ambiguities. The final alignment was 10,942 residues in length with ~61% average pairwise identity, with the LG+F+I+G4 model chosen as the best-fit model for phylogenetic inference. Alignments and tree files are given in *Figure 2—source data 2*.

## Repeat annotation and TE dynamics

TEs and other repeats were identified using the RepeatModeler and RepeatMasker pipelines. For each sample, a de novo repeat library was generated directly from the assembled nucleotides using RepeatModeler2 (*Flynn et al., 2020*) and combined with a database of 12,662 protostome repeats from Repbase v23.08 (*Bao et al., 2015*) and 278 additional TEs manually curated from the *A. vaga* genome (*Flot et al., 2013*). Repeats and TEs were then detected and classified using RepeatMasker v4.1.0 (*Smit et al., 2013*), and resultant outputs were post-processed using the 'One code to find them all' Perl script (*Bailly-Bechet et al., 2014*). The breakdown of TE superfamilies in the final database was 4145 DNA transposons (including 300 rolling circles), 5523 LTRs, 2583 LINEs (including SINEs), 227 PLEs, and 165 simple or low-complexity repeats. TE content (expressed as a proportion of genome size) was mapped onto the phylogeny using 'contMap' in the Phytools v0.6–99 package in R v3.6.0 (*R Development Core Team, 2016*; *Revell, 2012*). There is no module for the detection of class II MITEs in RepeatMasker; for these, the separate program Generic Repeat Finder (GRF) was run using default parameters. TE dynamics were investigated by constructing Kimura 2-parameter

divergence (*Kimura, 1980*) landscapes using the utility scripts in the RepeatMasker package and plotted using custom scripts (see below). Selected assemblies were also submitted to the REPET v2.5 'TEdenovo' (*Flutre et al., 2011*; *Quesneville et al., 2005*) TE detection and annotation pipeline with default parameters, for comparison. In addition, for *D. carnosus* and *R. sordida* reference assemblies, we increased the parameter 'minNbSeqPerGroup' from 3 to 5 to evaluate contribution from tetraploid genes, which was judged to be negligible. Although REPET *denovo* TE consensus sequences are automatically classified using Wicker's TE classification (*Wicker et al., 2007*), Repeat-Masker was additionally applied for further TE classification, detection, and landscape divergence plot building.

To provide a broad-brush comparison of TE content in rotifers relative to other animals, we applied the same RepeatMasker pipeline to 17 further animal genomes (see *Figure 2—source data 3* for further details). All genomes were downloaded from Ensembl Metazoa (https://metazoa. ensembl.org/index.html), except for *P. vanderplanki* and *P. nubifer* (http://bertone.nises-f.affrc.go. jp/midgebase/) and *R. varieornatus* and *H. exemplaris* (http://ensembl.tardigrades.org/index.html). It is important to note that this analysis is not intended as an exhaustive comparison of TE content evolution among protostome animals, but to provide points of reference using model systems more widely studied than rotifers.

## Mapping phylogenetic shifts in TE content

We reconstructed changes in the frequency of class I and class II TEs within the genome in turn onto phylogenetic trees as continuous characters. The null model assumed a single rate of evolution under a Brownian motion model across the whole tree. We then tested for a significant shift in TE content in bdelloids as a whole clade by fitting a two-rate model with a separate rate for the stem branch leading to bdelloids. Models were fitted using the 'brownie.lite()' function in the R package Phytools (*O'Meara et al., 2006*; *Revell, 2012*). The comparison was repeated for the major sub-classes of TEs in addition to the two main classes. Next, we tested for significant shifts in frequency on any branches across the tree using the Medusa-like approach implement in the 'transformPhylo.ML ()' function in the R package MOTMOT (*Thomas and Freckleton, 2012*), using the 'tm2' algorithm. In brief, this method searches all branches on the tree for a shift in evolutionary rate either localised to a single-branch or shared by all descendants (i.e. the whole clade), identified by calculation of stepwise AIC (Akaike Information Criterion). In order to focus on finding the major shifts in TE content on the tree, the number of permitted shifts were limited to two.

## Phylogenetic linear models for TE load

To assess differences in TE and repeat load between desiccating versus nondesiccating bdelloids, we ran Bayesian linear mixed-effects models of ($\log_e$) TE load (as a percentage of genome span) including desiccating/nondesiccating as a two-level fixed factor and sample ID as a random intercept term. The rotifer BUSCO gene phylogeny (shown in *Figure 2A*) was used to account for non-independence among species. A separate model was run for each TE/repeat classification (DNA transposons, rolling circles, PLEs, LTRs, LINEs, satellite, simple, low complexity, and unclassified repeats) to allow the pattern of TE load to vary across the phylogeny for different classifications. Inverse-Wishart priors were used for the random and residual variances, and models were run for 42,0000 iterations with a burn-in of 20,000 and a thinning interval of 200. This resulted in 2000 stored samples of the posterior with minimal autocorrelation (<0.2) in all cases (*Garamszegi, 2014*). Models were run using the MCMCglmm v2.29 (*Hadfield, 2010*) package in R. The phylogenetic signal, defined as the proportion of the total variance in TE content attributable to the phylogeny (*de Villemereuil and Nakagawa, 2014*), was estimated from the MCMCglmm model output using the formula: $\lambda = \sigma P^2/(\sigma P^2 + \sigma R^2)$. The same approach was used to test for an effect of desiccation on an expanded dataset of 59 protostome taxa, using the phylogeny provided in *Figure 2—source data 2*.

## Phylogenetic linear models for TE location and length

To assess differences in TE genomic location (i.e. the genomic context of TE insertions) between (1) monogononts versus bdelloids and (2) desiccating versus nondesiccating bdelloids, we counted the total span of three genomic features (other genes, other TEs and telomeric repeats) in a 50 kb window around a subset of class I TEs (those identified in *Figure 5A*) using BEDTools v2.29.2 'slop' (to

draw the window), 'intersect' (to find intersecting features) and 'groupby' (to count and summarise) (*Quinlan, 2014*; *Quinlan and Hall, 2010*). Other genes were defined as predicted genes that did not overlap with any TE annotation. Genomic locations of the telomeric hexamer 'TGTGGG' (*Gladyshev and Arkhipova, 2007*) were identified using EMBOSS 'fuzznuc' (*Rice et al., 2000*), excluding any hexamer that overlapped with a predicted gene. Note that the telomeric repeat for *Brachionus* is not known, but the sequence above was among the most frequent G-rich hexamers identified in the *B. calyciflorus* PSC1 genome. Phylogenetic linear models (as above) were then run with ($\log_e$) density of the three genomic features surrounding each focal TE as the response variable, either (1) monogonont/bdelloid or (2) desiccating/nondesiccating as a two-level fixed factor and sample ID as a random intercept term. Separate models were run for each TE classification (PLEs, LTRs and LINEs) using the same parameters specified above.

TE lengths of individual TEs were parsed directly from the final TE annotation (*Nowell et al., 2021*). Since this resulted in >1 million observations, TE lengths were averaged to the superfamily level using the classification system of *Kapitonov and Jurka, 2008* to assign superfamilies (i.e. trimming repeat names after the '#' character). Phylogenetic linear models (as above) were then run with ($\log_e$) length as the response variable, and with either (a) monogonont/bdelloid or (b) desiccating/nondesiccating as a two-level fixed factor. Sample ID was included as a random intercept term. Separate models were run for each TE classification (DNA transposons, PLEs, LTRs, and LINEs) using the same parameters specified above. Note that PLEs are largely absent in the sampled monogononts and were not tested when comparing monogononts versus bdelloids.

## LTR-R presence and absence

The presence or absence of specific LTR retrotransposon (LTR-R) insertions in our population data was inferred using a read-mapping approach. Specifically, the presence of a given insertion was scored based on the alignment score of the 'best' read that mapped continuously and contiguously across the LTR-genome boundary. First, full-length LTR-Rs (i.e. those with annotated 5' and 3' LTR regions) were identified from each reference assembly using LTR_retriever v2.8 (*Ou and Jiang, 2018*). Three filters were then applied to remove false positives. Candidates that showed an overlap with a predicted gene in the 5' or 3' LTR itself or an 'N' base within 150 bases upstream or downstream of its genomic location that might indicate local mis-assembly were removed. Candidates also required supporting evidence of LTR homology from a separate RepeatMasker annotation of the reference assembly. For each remaining LTR-R, a library of 'LTR-tags' was then generated by extracting a 100 bp sequence that spanned 50 bases into the genomic (i.e. non-TE) region of the insertion site from both the 5' and 3' terminal repeated regions. Thus, each pair of 'LTR-tags' represents an insertion of a particular LTR into a specific location in the focal genome, and a score is calculated based on the alignment information contained in the CIGAR string of the 'best' read (i.e. with the highest number of alignment matches) from the SAM mapping file: $S_i = ((M_{Li} - X_{Li}) + (M_{Ri} - X_{Ri}))/200$, where $M_{Li}$ is the number of alignment matches for the left-hand tag for LTR $i$, penalised by the number of mismatches $X_{Li}$, with equivalent scoring for the right-hand tag. Since tag length is 100 bases, the maximum score for a perfect alignment is 200, or one after normalisation. The number of mapped reads is also recorded to provide an estimate of coverage (but note that $S_i$ is taken from the best read only). Sequencing reads from all single-individual rotifer samples were aligned to the filtered LTR-tag set using BBTools 'bbmap' with the parameters 'minid=0.5 local=t' and scored using the above system. Because orthologous LTR-Rs may be identified from searches started in different genomes, we identified these cases by reconstructing the phylogeny of the LTR-tags and any with pairwise sequence divergence less than 0.1 were collapsed to yield a condensed final matrix.

The LTR-tag case-study in *Figure 4B* was selected for closer investigation in the draft assemblies after consideration of several examples, because it illustrates variability for an element insertion site within a species and indicates that Class I TEs can insert in coding regions, with potential fitness consequences. The LTR-tags were mapped to the RM15 draft assembly using Geneious, and were found to match an element annotated by LTR_retriever, containing four predicted genes. In the RM9 draft assembly, only the left-hand tag was mapped, as the scaffold ended before the inserted element was fully assembled. For the same reason, the element in RM9 had not been annotated as such by LTR_retriever, but the sequence is nearly identical (99.7%) to the insertion in RM15 along its aligned length (except that the annotations predicted three element-associated genes rather than four). The scaffolds were trimmed to the focal gene and aligned, and the region was used as a BLASTn query

against local databases for two other *R. magnacalcarata* reference genomes where the LTR-tag was absent: MAG1 and Rg2018. In each case, this provided the location of a closely similar but uninterrupted copy of the focal gene, although the annotations of the gene's structure differed slightly among genomes. These scaffolds were trimmed and aligned against the copies from RM9 and RM15, using the Geneious alignment tool with default settings, except that the gap extension penalty was reduced from 3 to 0.2 to enable the algorithm to handle the element insertion. Local features were manually reannotated to illustrate the interpretation provided in the text. To investigate the potential function of the interrupted gene, the copy from MAG1 (gene ID = g37061) was translated and used as a BLASTp query against the NCBI RefSeq Protein Database (*Pruitt et al., 2007*). A region of approximately 1000 residues was found to have weak similarity (~25% pairwise identity) to proteins annotated as midasins, from a range of eukaryotes. As a final step, the intact gene from MAG1 was used as a BLASTn query against the full draft genomes of RM9 and RM15, which revealed a separate scaffold in each case, containing a partial copy of the gene in which the coding sequence was intact across the junction spanned by the LTR-tag, and the element insertion was absent.

## TE transcription

Available RNA-seq libraries for *A. ricciae*, *A. vaga*, *R. magnacalcarata*, *R. socialis* and *R. sordida* were downloaded from the Sequence Read Archive (run accessions ERR2135448, SRR7962068, SRR2429147, SRR2430028, and SRR2430030) and mapped to the corresponding genome (Ar2018, Av2013, Rg_MAG1_maxhap, Rs_AK11_maxhap, and Rd_10x_p, respectively) using BBTools 'bbmap' with default parameters. The number of annotated TEs showing 100% coverage from mapped data (i.e. the genomic coordinates for the putative TE were completely covered by RNA-seq reads) was counted using BEDTools 'intersect' with the parameters '-c -f1' (*Quinlan, 2014*; *Quinlan and Hall, 2010*).

## Recombination analyses

We tested sexual versus clonal patterns of variation in LTR presence and absences. First, we calculated consistency indices (CI) with parsimony reconstruction of the binary matrix. LTR-tags with scores > 0.875 were coded as present (i.e. no more than half of the genome context or LTR region from both left and right LTR-tags was missing) and <0.875 coded as absent (alternative thresholds led to the same qualitative results). A CI = 1 indicates perfect nesting with no homoplasy, whereas a score less than one is expected if variation is shuffled among loci and not tree-like. Next, we calculated the index of association and ran permutations to test for significant linkage disequilibrium of the LTR-tag data relative to a null model of random shuffling (expected in a fully outcrossing sexual population). We used the modified index of association by *Agapow and Burt, 2001*, that corrects for an effect of the number of loci on the index, and ran permutations using the 'ia' function in the Poppr v2.8.5 library (*Kamvar et al., 2014*) in R. Data were coded as diploid and codominant presence/absence data (because of the lack of diploid assemblies in the population-level data). Finally, for *R. magnacalcarata* and *R. socialis* we ran simulations with the FacSexCoalescent simulator of *Hartfield et al., 2018*, to generate 50,000 datasets with the same number of individuals and sampled binary loci as observed, but with frequencies of sexual versus asexual reproduction within the populations varying from $10^{-7}$ (i.e. negligible) to 1 (i.e. obligate sexual). We estimate the posterior distribution of the frequency of sex for our observed samples using Approximate Bayesian Computation on the simulated datasets implemented in the 'abc' package in R. The simple 'rejection' algorithm was used, accepting parameter values yielding simulated metrics within a Euclidean distance of 0.05 from the observed values. The simulation model assumes constant population size and constant transposition rate across individuals, together with the other usual neutral coalescent assumptions. However, the metrics used for recombination are statistical rather than model based, and hence the main limitation to the test's power is whether conditions permit sufficient polymorphism of TE presence/absence in multiple insertion sites to detect statistical associations between loci or not.

## Reverse transcriptase survey

A hidden Markov model (HMM) approach was used to survey the predicted rotifer proteomes for proteins encoding the reverse transcriptase (RT) domain (Pfam ID PF00078). First, a HMM was constructed from an alignment of 51 RT domains from across the tree of life (*Arkhipova et al., 2003*), supplemented with 67 bdelloid-specific retroelements (*Arkhipova et al., 2003*; *Flot et al., 2013*; *Gladyshev et al., 2007*; *Gladyshev and Arkhipova, 2010b*; *Gladyshev and Arkhipova, 2009*; *Gladyshev and Arkhipova, 2007*). Alternative transcripts were first removed from predicted proteomes and proteins with a significant match (*E*-value $\leq$1e−5) were identified and inserted into the core RT alignment using HMMER v3.2.1 'hmmsearch' and 'hmmalign', respectively (http://hmmer.org/). Maximum likelihood phylogenies were then constructed using IQ-TREE as above, specifying the root of the phylogeny to be on the branch leading to the bacterial retrons (*Arkhipova et al., 2003*). Trees were manipulated using FigTree v1.4.4 (*Rambaut, 2007*), colouring the identified RT-encoding rotifer proteins based on their phylogenetic position.

## RNAi pathways survey

A similar HMM based approach was used to evaluate copy-number evolution of three key pathways involved in RNAi gene-silencing. Putative Argonaute proteins were identified based on the presence of both the PAZ and PIWI domains (Pfam IDs PF02170 and PF02171 respectively), putative Dicer proteins were identified based on the presence of both PAZ and Dicer (PF03368) domains, and putative RdRP proteins were identified based on the presence of the RdRP domain (PF05183). Stockholm files were downloaded from Pfam (*El-Gebali et al., 2019*) and aligned to the proteomes using HMMER (*E*-value $\leq$1e−5) as above. Reference proteomes from a selection of eukaryotic species to represent the diversity and distribution of Argonaute, Dicer and RdRP proteins were downloaded (June 2020) from UniProt and subjected to the same procedure: *Arabidopsis thaliana* (UP000006548), *Oryza sativa* (UP000007015), *Neurospora crassa* (UP000001805), *Schizosaccharomyces pombe* (UP000002485), *Laccaria bicolor* (UP000001194), *Dictyostelium discoideum* (UP000002195), *D. melanogaster* (UP000000803), *C. elegans* (UP000001940), *H. exemplaris* (UP000192578), *H. robusta* (UP000015101), *L. gigantea* (UP000030746), *S. haematobium* (UP000054474), *B. plicatilis* (UP000276133), *Branchiostoma floridae* (UP000001554), and *Homo sapiens* UP000005640. The evolution of bdelloid RdRP was further analysed using a core alignment of 538 RdRP sequences (247 eukaryotic species) from a recent study by *Pinzón et al., 2019*. Proteins were aligned using either 'hmmalign' from the HMMER package or Clustalo, and ML phylogenies were constructed using IQ-TREE as above.

## RNAi gene-family expansion analyses

To test whether the number of RNAi genes found in bdelloid genomes was expanded relative to other eukaryotes, we used the CAFE software, which fits a constant birth rate model to predict gene family size evolution to a phylogenetic tree (*Hahn et al., 2005*; *Han et al., 2013*). We repeated the analysis fitting both a single and separate birth rate for each gene family, Ago, Dicer and RdRP. The method then tests for significant shifts in copy number on each branch on the phylogenetic tree. To account for the large number of p-values (one for each branch), we calculated corrected p-values using the False Discovery Rate method (*Benjamini and Hochberg, 1995*) and reported significant changes occurring at the level of class or above in the tree of syndermatans and the protostome outgroups.

To explore the impact of ploidy differences on comparisons, we constructed a 'comparative protein–domain abundance' plot using counts of Pfam domain entries parsed directly from InterProScan5 (*Jones et al., 2014*) annotation of predicted proteomes. For each Pfam domain, an 'abundance score' was computed as the ($\log_e$) ratio of domain counts in bdelloids divided by the domain counts in eukaryotes, corrected for inflation in bdelloids due to the ancient whole-genome duplication by dividing the former by two. This simple correction is likely to be conservative, since many loci are once again diploid, having lost one branch of the ancient duplication (i.e. tetraploidy is degenerate). Thus, the abundance ratio for a given domain provides an approximate measure of the number of gene copies in bdelloids relative to the average number of gene copies in eukaryotes (acknowledging that our 'eukaryote' sample here is not exhaustive but restricted to the species listed above). An abundance ratio >0 indicates a greater number of copies (on average) in bdelloids

relative to eukaryotes. To check that the putative RdRP expansion was indeed eukaryotic in origin, rather than viral, the HMMs for four viral RdRP families (PF00680, PF00978, PF00998, and PF02123) were downloaded from Pfam and submitted to the same search protocol, with zero hits to bdelloid proteomes recorded.

## Code availability

All TE analysis scripts used in this study are available at https://github.com/reubwn/te-evolution; *Zoni, 2021*; copy archived at swh:1:rev:68693e5a4368a604f8eaaa693f9436a0376ca3a8.

## Acknowledgements

Genome sequencing was performed by the UK Natural Environment Research Council (NERC) Bio-molecular Analysis Facility at the Centre for Genomic Research (CGR) at the University of Liverpool (NBAF-Liverpool) and the DNA Sequencing Facility in the Biochemistry Department at the University of Cambridge. The authors wish to thank the following: Christiane Hertz-Fowler, Pia Koldkjær and John Kenny (CGR), Shilo Dickens and Nataliya Scott (Cambridge). Matthew Arno and Colin Sharp (Edinburgh), and Steven Van Belleghem (University of Puerto Rico) for support with the planning and execution of various aspects of genome sequencing and/or assembly, Tom Smith and Anita Kristiansen for rotifer sampling, Mike Tristem for helpful discussions on detecting LTR polymorphisms, Hannah Froy for help with phylogenetic modelling, and Julie Blommaert and two anonymous reviewers for critical feedback on a previous version of this manuscript.

## Additional information

### Funding

| Funder | Grant reference number | Author |
| --- | --- | --- |
| Natural Environment Research Council | NE/M01651X/1 | Timothy G Barraclough |
| Natural Environment Research Council | NE/S010866/2 | Christopher G Wilson<br>Timothy G Barraclough |

The funders had no role in study design, data collection and interpretation, or the decision to submit the work for publication.

### Author contributions

Reuben W Nowell, Conceptualization, Resources, Data curation, Software, Formal analysis, Validation, Investigation, Visualization, Methodology, Writing - original draft, Project administration, Writing - review and editing; Christopher G Wilson, Conceptualization, Formal analysis, Supervision, Validation, Investigation, Writing - original draft, Project administration, Writing - review and editing; Pedro Almeida, Philipp H Schiffer, Diego Fontaneto, Lutz Becks, Resources, Data curation, Writing - review and editing; Fernando Rodriguez, Software, Formal analysis, Validation, Methodology, Writing - review and editing; Irina R Arkhipova, Formal analysis, Validation, Investigation, Methodology, Writing - review and editing; Timothy G Barraclough, Conceptualization, Resources, Formal analysis, Supervision, Funding acquisition, Validation, Investigation, Visualization, Methodology, Writing - original draft, Project administration, Writing - review and editing

### Author ORCIDs

Reuben W Nowell (iD) https://orcid.org/0000-0001-7546-6495
Philipp H Schiffer (iD) http://orcid.org/0000-0001-6776-0934
Irina R Arkhipova (iD) http://orcid.org/0000-0002-4805-1339

### Decision letter and Author response

Decision letter https://doi.org/10.7554/eLife.63194.sa1
Author response https://doi.org/10.7554/eLife.63194.sa2

# Additional files

## Supplementary files
• Transparent reporting form

## Data availability
All raw sequencing data have been deposited in the relevant International Nucleotide Sequence Database Collaboration (INSDC) databases under the Study ID PRJEB43248. Genome assemblies and gene predictions have been deposited at DDBJ/ENA/GenBank with the same Study ID. Figure 2 source data (RepeatMasker output files) has been uploaded to Dryad Digital Repository (https://doi.org/10.5061/dryad.fbg79cnsr).

The following datasets were generated:

| Author(s) | Year | Dataset title | Dataset URL | Database and Identifier |
|---|---|---|---|---|
| Nowell RW, Wilson CG, Almeida P, Schiffer PH, Fontaneto D, Becks L, Rodriguez F, Arkhipova IR, Barraclough TG | 2021 | RepeatModeler and RepeatMasker output files | https://doi.org/10.5061/dryad.fbg79cnsr | Dryad Digital Repository, 10.5061/dryad.fbg79cnsr |
| Nowell RW, Wilson CG, Almeida P, Schiffer PH, Fontaneto D, Becks L, Rodriguez F, Arkhipova IR, Barraclough TG | 2021 | Evolutionary dynamics of transposable elements in bdelloid rotifers: Umbrella project for PRJEB43238 and PRJEB43239 | https://www.ebi.ac.uk/ena/browser/view/PRJEB43248 | European Nucleotide Archive (ENA), PRJEB43248 |

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
