## [Decision Letter]

**Acceptance summary:**

Nowell et al. present an analysis of transposable elements in bdelloid rotifers, which are thought to be an ancient asexual lineage, and compare their dynamics to those in related species that have retained sexual reproduction. Through this comparative analysis, the authors test various evolutionary hypotheses about asexual genomes, as well as recent suggestions that bdelloid rotifers may not actually be asexual. The authors find no evidence supporting the presence of sex in bdelloid rotifers, however, the dynamics of transposable elements are inconsistent with evolutionary theory predictions. This inconsistency may be explained by the authors' finding of an expansion of RNAi-related genes, which may play a role in countering the expected TE dynamics in asexual species. Overall, this work is substantial, thorough, and presents some answers to long-standing questions about the genome evolution of long-term asexual species.

**Decision letter after peer review:**

Thank you for submitting your article "Evolutionary dynamics of transposable elements in bdelloid rotifers" for consideration by *eLife*. Your article has been reviewed by three peer reviewers, and the evaluation has been overseen by a Reviewing Editor and Patricia Wittkopp as the Senior Editor. The following individual involved in review of your submission has agreed to reveal their identity: Julie Blommaert (Reviewer #3).

The reviewers have discussed the reviews with one another and the Reviewing Editor has drafted this decision to help you prepare a revised submission.

We would like to draw your attention to changes in our revision policy that we have made in response to COVID-19 (https://elifesciences.org/articles/57162). Specifically, we are asking editors to accept without delay manuscripts that they judge can stand as *eLife* papers without additional data, even if they feel that they would make the manuscript stronger.

Summary:

Nowell et al. present an analysis of transposable elements (TEs) in bdelloid rotifers and compare their dynamics to those in related species. Through this comparative analysis, the authors test various evolutionary hypotheses about asexual genomes, as well as recent suggestions that these ancient asexual organisms may not actually be asexual. Nowell et al. find no evidence supporting the presence of recombination (and thus, sex) in bdelloid rotifers, and no strong predicted evolutionary signatures of asexuality in TE dynamics in these species. Additionally, they find evidence for expansion of RNAi-related genes, which may play a role in countering the expected TE dynamics in asexual species. Overall, this work is substantial, thorough, and presents some answers to long-standing questions about the genome evolution of long-term asexual species.

Revisions:

1) There is a comparison in the manuscript between Bdelloids and Monogonants. It wasn't clear however that these groups had been sampled sufficiently. The Monogonants are represented by 5 species (8 genomes) within a single genus in no way representing the diversity of Monogonants and the sampling of Bdelloids is also small. The authors should take a more cautious tone to any conclusions.

2) The rationale for focusing on this specific group of TEs did not appear robust. The authors say "this class of TEs is thought to be least likely to undergo horizontal transfer and thus the most dependent on sex for transmission". But other groups are not evolving predominantly by horizontal transfer, transposons can change without meiotic sex and this section needs writing a little more clearly. The following lines make a case that some transposon groups increase, and some decrease in frequency. The obvious hypothesis is drift, but the writing was unclear, and it always felt that some other mechanism was being proposed but never really stated clearly.

3) Subsection “Abundant and diverse TEs in bdelloid genomes”, comparison of TE abundances across animals; this section was very poorly done. The authors could delete this comparison and have a better manuscript. How were these other species chosen? Is *C. elegans* a good representative of the entire phylum Nematoda? Are the tardigrades representatives of their phylum? Assembly and annotation methods vary enormously across datasets so what can the authors conclude without standardizing assembly and annotation for these other animal groups? The authors say "as expected, both the abundance and diversity of TEs varied widely across taxa" This was indeed expected, Figure 2B seems to show noise, and suggests that the inclusion of this data was not a good idea. we suggest it is removed, or a very substantive analysis and discussion of the way in which it is an accurate and representative sample of animal transposon loads is written.

4) The authors need to make it very clear that this is not a test, it is a single observation. The phrase "as predicted by theory for elements dependent on vertical transmission" seems rather unsupported. Does this relate to the argument put forward in the third paragraph of subsection “Abundant and diverse TEs in bdelloid genomes”? It was unconvincing at this point also. The current description that some families increase and some decrease is couched in what sounds like too meaningful sounding language, which could be improved to be more consistent with the results. Paragraph five of that section seem to make an argument that the variation of TEs in bdelloids is purely a phylogenetic effect variably present in some bdelloid lineages and related groups. If this is their view (and it seems very reasonable indeed) then the manuscript would be improved considerably if they stated it more clearly.

5) "Consistent with a high fit of the data to the phylogeny under a Brownian motion model as would be expected if TE load evolves neutrally along branches of the phylogeny." This was a truly excellent result that needed to be put forward more strongly in other areas of the manuscript. In this area, and some others in this manuscript the authors have truly unique data dramatically improving our understanding of bdelloids. The manuscript would be improved if authors concentrated much less throughout on ideas, since these data are exceptional and different from other animals, and instead followed their own analysis that this fits with current biological thought.

6) "No significant difference between monogononts and bdelloids, or between desiccating and non-desiccating bdelloids" It is not clear what statistical test is being carried out. All tests require phylogenetic control of course. We do agree that they are quite similar, perhaps this should be rewritten to reflect only that?

7) The authors look at 3 gene families concerned with transposon control to examine copy number. In one of them they say "the RdRP domain in particular is significantly expanded". It is unclear what test of significance was carried out and where to find this analysis. Unlike the query concerning desiccating and non-desiccating above, this analysis is essential. The authors make a really big thing about the expansion of this gene family, including it in the Abstract. If they wish to keep its prominence then they need to clearly show whether there is evidence that the size of this domain family is significantly expanded along the branch leading to bdelloids. We understand that this is illustrated in Figure 7 but this is not a test. This needs to be made much clearer in a quantitative rather than descriptive way. There is need for broad taxonomic sampling, standardization of assembly and annotation, and a phylogenetic design for this analysis. Else it should be removed or at the least described more conservatively.

8) "Why do bdelloids possess such a marked expansion of gene silencing machinery?" There is no evidence presented that they do. There may be a hypothesis that they do it differently, rather than more, but that also needs testing. There is a lot of speculation in this paragraph – removing it altogether would improve the manuscript.

9) If there is an expansion of this family what can we then conclude? The authors say in the Abstract "bdelloids share a large and unusual expansion of genes involved in RNAi-mediated TE suppression. This suggests that enhanced cellular defence mechanisms might mitigate the deleterious effects of active TEs and compensate for the consequences of long-term asexuality" yet they also review that animal groups can utilize different gene families for transposon control. Is there evidence that clade 5 nematodes with PIWI have a quantitatively different transposon defence mechanism? No, they just use a different pathway to some other groups, and the default position surely has to be the same for bdelloids, there is no evidence presented that their defence is enhanced. We strongly recommend that the authors reduce the strength of their claims about the significance of bdelloid transposon control gene families in this manuscript.

10) The Conclusions (and Abstract) were too speculative and not fully supported by the existing data, though this can easily be addressed by a substantial re-write.

11) Regarding hypothesis 4, the authors test whether or not desiccating species have lower TE loads than non-desiccating species, but the logic outlined in paragraph seven of the Introduction suggests that the relationship between desiccation and TE load may be more nuanced that overall TE load. It could be possible that DSB repair associated with desiccation removes only recent insertions if homologous pairing is involved, or high-copy TEs if ectopic recombination has occurred. The authors already test recent TE activity elsewhere in the manuscript, so they could compare signatures of recent activity in desiccating vs non-desiccating species to see if there are fewer recently active TEs in desiccation species. Similar comparisons could easily be made for abundance of high-copy TEs (regardless of length).

12) Additionally, regarding the signatures of recent transposition, the authors have done a thorough job comparing TE divergences and LTR insertions, but since transcriptomes for some species are available, presence of transcribed TEs could provide further support for recent and ongoing TE activity.

13) Materials and methods section, please provide more details about the part carried with the abc package itself (e.g. which regression/rejection algorithms were used, etc.).

14) Horizontal gene transfer: given the abundance of recent DNA transposons in some clades (class I), it may be worth discussing a bit more this possibility (at this stage it is mostly discussed in the Conclusion).

15) If we understand correctly, there is no assessment of TEs or SNPs heterozygosity for each individual. This might be interesting to explore. If TEs are deleterious recessive, one might observe more frequently at the heterozygous state. For intraspecific data, it may be interesting to look at how nucleotide diversity varies along the genome. Since variable recombination may be associated with diversity due to the effects of selection at linked sites, checking diversity along the genome may bring another layer of information about the frequency of sexual reproduction and its effects on TEs diversity. Although this would be a rather exploratory analysis it would be interesting to know how do methods designed to estimate effective recombination rates perform on these data (e.g. LDHat, or more recently iSMC for a single diploid genome).

16) Question related to demography and selection: would it be possible to obtain estimates of the effective population size for these clades? It would be interesting to have such an estimate to get an idea of the efficiency of purifying selection against TEs, and whether Muller's ratchet could explain the current abundance of TEs (in the case of moderate/small effective population sizes).

17) We liked the idea of using the ABC to test for consistency with asexuality, but to what extent is it biased by non-constant transposition rates, which cannot be properly modeled by the coalescent simulation? One would also assume these simulations do not take into account past changes in demography? This is not necessarily a major issue, as long as these limitations are mentioned.

---

## [Author Response]

Revisions:1) There is a comparison in the manuscript between Bdelloids and Monogonants. It wasn't clear however that these groups had been sampled sufficiently. The Monogonants are represented by 5 species (8 genomes) within a single genus in no way representing the diversity of Monogonants and the sampling of Bdelloids is also small. The authors should take a more cautious tone to any conclusions.

Our analysis includes all publicly available genome sequences for both bdelloids and monogononts. However, we agree that both groups comprise a huge diversity of genera, most of which remains currently unsampled. We now state explicitly that our comparison is not exhaustive. Subsequent references to “bdelloid” versus “monogonont” comparisons and analyses have been edited to clarify that they refer only to comparison of our sampled genomes.

2) The rationale for focusing on this specific group of TEs did not appear robust. The authors say "this class of TEs is thought to be least likely to undergo horizontal transfer and thus the most dependent on sex for transmission". But other groups are not evolving predominantly by horizontal transfer, transposons can change without meiotic sex and this section needs writing a little more clearly. The following lines make a case that some transposon groups increase, and some decrease in frequency. The obvious hypothesis is drift, but the writing was unclear, and it always felt that some other mechanism was being proposed but never really stated clearly.

We agree that the comparison of class I and class II TEs was not presented clearly in the text, and we have made edits to clarify throughout. The key point is that previous work finds that certain class I TEs, namely the LINE-like superfamily, are almost exclusively vertically transmitted, whereas some class II TEs are maintained by horizontal transfer or beneficial effects. For this reason, previous authors have predicted that certain class I TEs are more dependent, on average, on sexual reproduction for their maintenance than class II TEs. The predicted effects of the loss of sex should therefore be more pronounced for class I than class II TEs, and particularly so for LINEs. We have added this as a clearer prediction in our general introduction, emphasized previous work addressing this idea for class I and class II TEs in bdelloids, added it to the outline of our goals, and reworded the Results section to reflect these changes. We hope this point is now clear. As the reviewers recommend, we now make more explicit reference to the drift hypothesis in both the analyses and conclusions.

3) Subsection “Abundant and diverse TEs in bdelloid genomes”, comparison of TE abundances across animals; this section was very poorly done. The authors could delete this comparison and have a better manuscript. How were these other species chosen? Is C. elegans a good representative of the entire phylum Nematoda? Are the tardigrades representatives of their phylum? Assembly and annotation methods vary enormously across datasets so what can the authors conclude without standardizing assembly and annotation for these other animal groups? The authors say "as expected, both the abundance and diversity of TEs varied widely across taxa" This was indeed expected, Figure 2B seems to show noise, and suggests that the inclusion of this data was not a good idea. we suggest it is removed, or a very substantive analysis and discussion of the way in which it is an accurate and representative sample of animal transposon loads is written.

The section was not intended to be an exhaustive analysis of TE content evolution across protostome animals more generally, and we agree that it cannot be presented as such. Instead, these taxa are included as representative outgroups for interpreting bdelloid TE evolution in a phylogenetic context. As the reviewers point out, the data ignore variation in TE content within each phylum (e.g., for Nematoda), as well as other potential issues caused by genome assembly method/quality etc. We have edited and shortened this section to make clearer the purpose of the comparison, while drawing attention to its limitations. This is reiterated in the Materials and methods section, with further details on the sampling of taxa.

To explain further, our goals for including other animal genomes are threefold: (1) To provide context for the results in bdelloids by showing the repeat content in a selection of other animal genomes, including well-known model species such as *D. melanogaster* and *C. elegans*, and a selection of other protostome genomes that were chosen from Ensembl Metazoa (and might be considered “benchmark” genomes). In addition, we included the very recently published genome of the acanthocephalan *Pomphorhynchus laevis* (Mauer et al., 2020), a close relative to bdelloids and monogononts. (2) To emphasise the fact that, contrary to previous ideas put forward in the literature, bdelloids do not encode particularly low proportion of TEs, as is clear even when compared to this small set. It would be impossible to make this important point without comparison to other animal genomes. (3) To compare two further pairs of desiccating and nondesiccating species in two more distantly related taxa: tardigrades (*Ramazzottius varieornatus* and *Hypsibius exemplaris*) and flies (*Polypedilum vanderplanki* and *P. nubifer*), which again provides useful context for the bdelloid result.

In light of these, we believe the figure panel makes a small but important and necessary contribution to the study: to provide visual context for interpreting TE content in bdelloids, alongside a small selection of other taxa and well-known model species that readers will be familiar with.

4) The authors need to make it very clear that this is not a test, it is a single observation. The phrase "as predicted by theory for elements dependent on vertical transmission" seems rather unsupported. Does this relate to the argument put forward in the third paragraph of subsection “Abundant and diverse TEs in bdelloid genomes”? It was unconvincing at this point also. The current description that some families increase and some decrease is couched in what sounds like too meaningful sounding language, which could be improved to be more consistent with the results. Paragraph five of that section seem to make an argument that the variation of TEs in bdelloids is purely a phylogenetic effect variably present in some bdelloid lineages and related groups. If this is their view (and it seems very reasonable indeed) then the manuscript would be improved considerably if they stated it more clearly.

The reviewers rightly highlight a challenge in understanding bdelloid rotifer evolution, namely that their origin is a single evolutionary event. Nonetheless, we can test whether shifts are associated with this particular branch or with deeper branches on the tree. If they are not found on this branch, we can say there is no evidence consistent with the asexuality hypothesis, which predicts that the loss of sex should have a major effect on TE content. If a major shift is found (evaluated against a null model of background rates of evolution), we can say it is consistent with the asexuality hypothesis (though of course we could not rule out other explanations for that change based on a single event). We have clarified our hypotheses and predictions in the Introduction, added new phylogenetic statistical tests in places that were lacking a clear test, and clarified throughout the limitations raised by the reviewer.

Yes, the statement “as predicted by theory for elements dependent on vertical transmission” does relate to the argument put forward in the previous version of the manuscript. We have now edited all of these sections and we hope this has improved the overall clarity of the argument being made. We have also added further references in all sections to support the statement. Please see the response to comment #2 also.

5) "Consistent with a high fit of the data to the phylogeny under a Brownian motion model as would be expected if TE load evolves neutrally along branches of the phylogeny." This was a truly excellent result that needed to be put forward more strongly in other areas of the manuscript. In this area, and some others in this manuscript the authors have truly unique data dramatically improving our understanding of bdelloids.

We thank the reviewer for their positive appraisal of this result. We have edited/added to various sections of the manuscript to try and make this result more prominent and integrated more consistently with the various other hypotheses.

The manuscript would be improved if authors concentrated much less throughout on ideas, since these data are exceptional and different from other animals, and instead followed their own analysis that this fits with current biological thought.

We disagree with the suggestion of the reviewer to remove reference to general ideas about sex and TEs or to hypotheses previously developed for bdelloids specifically. Our goal with this study was very much to test those well-established hypotheses using extensive new comparative data. Without this connection, it would be hard for readers to see how our results change previous thinking or impact the broader field. We have, however, added new material that we hope clarifies the background and explains exactly how each analysis relates to a particular hypothesis. We have also clarified our two main hypotheses—the asexuality and desiccation hypotheses—and separated them from the more exploratory analyses we perform to look at other possible effects on bdelloid TE evolution.

6) "No significant difference between monogononts and bdelloids, or between desiccating and non-desiccating bdelloids" It is not clear what statistical test is being carried out. All tests require phylogenetic control of course. We do agree that they are quite similar, perhaps this should be rewritten to reflect only that?

Statistical analyses throughout in the manuscript have now been standardised to use the same phylogenetic mixed-effects linear modelling approach (i.e., MCMCglmm). Thus, all statistical analyses now have appropriate phylogenetic controls, as the reviewer suggests. We have edited the text to make clear which statistical test was performed, while further details have been added to the relevant sections of the Materials and methods to better explain exactly what was done.

7) The authors look at 3 gene families concerned with transposon control to examine copy number. In one of them they say "the RdRP domain in particular is significantly expanded". It is unclear what test of significance was carried out and where to find this analysis. Unlike the query concerning desiccating and non-desiccating above, this analysis is essential. The authors make a really big thing about the expansion of this gene family, including it in the Abstract. If they wish to keep its prominence then they need to clearly show whether there is evidence that the size of this domain family is significantly expanded along the branch leading to bdelloids. We understand that this is illustrated in Figure 7 but this is not a test. This needs to be made much clearer in a quantitative rather than descriptive way. There is need for broad taxonomic sampling, standardization of assembly and annotation, and a phylogenetic design for this analysis. Else it should be removed or at the least described more conservatively.

As suggested, we have added a quantitative analysis of RdRP family size for our sample of genomes. Our test applies a birth-death rate model implemented in CAFE5, which calculates the significance of changes in family size along each branch on the phylogenetic tree. There is a highly significant shift towards larger family size on the stem branch of bdelloids. Details of this test are now added in Figure 8 and Materials and methods section.

We agree, however, that the detailed outcomes of this analysis depend on the taxonomic sampling. To address this limitation at least for RdRP, we have included an additional analysis that uses data from a thorough and recent study of RdRP evolution for ~250 eukaryote species, including plants and protists (Pinzón et al., 2019). The inclusion of the bdelloid RdRP copies into this broad taxonomic sample supports the conclusions that RdRP is substantially expanded in bdelloids, and that the majority of the bdelloid RdRPs are highly divergent from known RdRPs (see Figure 7—figure supplement 2). Nonetheless, we have also edited the RNAi section of the manuscript to draw out the most conservative conclusions from these comparisons. The key points are that bdelloids contain at least 5 times as many RdRP copies as has been observed in any other metazoan to date (including our sample). While it remains possible that other metazoan lineages exist with similar copy numbers bdelloids, none have been observed to date.

8) "Why do bdelloids possess such a marked expansion of gene silencing machinery?" There is no evidence presented that they do. There may be a hypothesis that they do it differently, rather than more, but that also needs testing. There is a lot of speculation in this paragraph – removing it altogether would improve the manuscript.

As described in the previous comment, we now present clear statistical and comparative evidence for an expansion in the RNAi machinery, and for RdRP in particular. We also note comment #29, which asked for *more* discussion of possible alternative explanations for the RNAi diversification. In agreement with that reviewer, we do think this result is an important component of the work and as such deserves at least some discussion of possible factors that might explain it. However, as suggested here, we have reduced the section about possible consequences and interpretations of our finding and moved it to the conclusions to emphasise that it is speculative. We have also added some text to highlight the alternative hypothesis indicated by the reviewer—that the RNAi system observed in bdelloids does not represent an “enhancement” against TEs but is simply an alternative set of pathways.

9) If there is an expansion of this family what can we then conclude? The authors say in the Abstract "bdelloids share a large and unusual expansion of genes involved in RNAi-mediated TE suppression. This suggests that enhanced cellular defence mechanisms might mitigate the deleterious effects of active TEs and compensate for the consequences of long-term asexuality" yet they also review that animal groups can utilize different gene families for transposon control. Is there evidence that clade 5 nematodes with PIWI have a quantitatively different transposon defence mechanism? No, they just use a different pathway to some other groups, and the default position surely has to be the same for bdelloids, there is no evidence presented that their defence is enhanced. We strongly recommend that the authors reduce the strength of their claims about the significance of bdelloid transposon control gene families in this manuscript.

We have made edits throughout the manuscript, including the Abstract and conclusions, to be more conservative with our conclusions regarding the biological significance of the detected RNAi diversification in bdelloids. We agree that it is possible that these RNAi pathways are simply being used in slightly non-canonical ways, and one possibility with respect to RdRP is discussed in the text.

10) The Conclusions (and Abstract) were too speculative and not fully supported by the existing data, though this can easily be addressed by a substantial re-write.

We have extensively edited both the Abstract and conclusions sections to remove or otherwise make clear which parts are supported by our data and which parts are proposed explanations. Please see our replies to comments #5, #7–10 and #28 which also relate to this comment.

11) Regarding hypothesis 4, the authors test whether or not desiccating species have lower TE loads than non-desiccating species, but the logic outlined in paragraph seven of the Introduction suggests that the relationship between desiccation and TE load may be more nuanced that overall TE load. It could be possible that DSB repair associated with desiccation removes only recent insertions if homologous pairing is involved, or high-copy TEs if ectopic recombination has occurred. The authors already test recent TE activity elsewhere in the manuscript, so they could compare signatures of recent activity in desiccating vs non-desiccating species to see if there are fewer recently active TEs in desiccation species. Similar comparisons could easily be made for abundance of high-copy TEs (regardless of length).

We agree that the relationship between TE load and anhydrobiosis may be more complex than the relatively simple predictions made in some of the literature. Our conclusions emphasises the reviewer’s point that “the relationship between desiccation and TE load may be more nuanced” — we state that the effects of DSB repair on TE load may in fact work both ways. In addition, it is possible that some mechanism of DSB repair, or other kinds of intragenomic recombination, are at work in nondesiccating species too. We also consider that such nuances may not be detectable with the current data (i.e., genomes in fragmented “draft” status), but may come to light in the future if multiple highly contiguous genomes from within populations/species become available.

If we understand correctly, the reviewer asks more specifically about two scenarios: whether desiccation-related DSB-repair might depend on (1) the age of the TE insertion (“DSB repair associated with desiccation removes only recent insertions if homologous pairing is involved”), or (2) copy-number of the TE (“or high-copy TEs if ectopic recombination has occurred”). We think the manuscript already presents data to test these two scenarios.

First, we tested the hypothesis that “DSB repair associated with desiccation removes only … highcopy TEs if ectopic recombination has occurred” using the phylogenetic mixed-effects modelling approach described in Figure 6B and Figure 6—figure supplement 2. Note that TE abundance *and* length are important factors in the ER model, since the rate of ER is negligible below a certain sequence length (found to be ~1 kb in mice, for example). We did not find any significant differences between desiccating and nondesiccating species with regard to the relationship between TE abundance and length.

Our data also address the hypothesis that “DSB repair associated with desiccation removes only recent insertions if homologous pairing is involved”. Specifically, we do not detect any consistent differences in the load of recent TEs between desiccating and nondesiccating species in our TE landscape plots (Figure 3 and Figure 3—figure supplements 1–7), which show evidence of recent activity in both desiccating and nondesiccating bdelloid species, as demonstrated by peaks in the low divergence bins to the left-hand side. This is most pronounced in the *Adineta* species but can also be seen in both desiccating and nondesiccating *Rotaria* lineages (e.g., Figure 3C, D and Figure 3—figure supplement 7). These plots show the proportion of the genome accounted for by each TE family (colour-coded in the plots) – thus, they might be considered a proxy for TE abundance regardless of length, as the reviewer suggests. Thus, the hypothesis that “DSB repair associated with desiccation removes only recent insertions if homologous pairing is involved” is not supported by the current data.

12) Additionally, regarding the signatures of recent transposition, the authors have done a thorough job comparing TE divergences and LTR insertions, but since transcriptomes for some species are available, presence of transcribed TEs could provide further support for recent and ongoing TE activity.

This is a nice idea, thank you. We have conducted a small supplementary analysis to ascertain what proportion of putative TEs show evidence of transcription by mapping publicly available RNA-seq data for *A. ricciae, A. vaga, R. magnacalcarata, R. socialis* and *R. sordida* onto their genome assemblies (Ar2018, Av2013, MAG1, AK11 and RSOR408, respectively), and calculating the proportion of TEs that show 100% coverage of RNA-seq reads across the genomic coordinates for the annotated TE. Results are variable across species and TE superfamilies, but ca. 25% putative TEs show at least some evidence of active transcription as defined above. We have included these results in the revised manuscript to further support the hypothesis that TEs are currently active in bdelloid genomes.

13) Materials and methods section, please provide more details about the part carried with the abc package itself (e.g. which regression/rejection algorithms were used, etc.).

We have added the following detail: “The simple “rejection” algorithm was used, accepting parameter values yielding simulated metrics within a Euclidean distance of 0.05 from the observed values” in the revised manuscript.

14) Horizontal gene transfer: given the abundance of recent DNA transposons in some clades (class I), it may be worth discussing a bit more this possibility (at this stage it is mostly discussed in the Conclusion).

Thanks for the suggestion. We have added a sentence to the Introduction to make this point more explicitly. Please see also the response to comments #2 and #4.

15) If we understand correctly, there is no assessment of TEs or SNPs heterozygosity for each individual. This might be interesting to explore. If TEs are deleterious recessive, one might observe more frequently at the heterozygous state. For intraspecific data, it may be interesting to look at how nucleotide diversity varies along the genome. Since variable recombination may be associated with diversity due to the effects of selection at linked sites, checking diversity along the genome may bring another layer of information about the frequency of sexual reproduction and its effects on TEs diversity. Although this would be a rather exploratory analysis it would be interesting to know how do methods designed to estimate effective recombination rates perform on these data (e.g. LDHat, or more recently iSMC for a single diploid genome).

In the case highlighted in Figure 4B, we assessed TE heterozygosity and discussed the evidence for a deleterious recessive effect and its relationship to bdelloid genome structure. However, the reviewer is correct that we do not assess patterns of SNP diversity or TE heterozygosity more broadly in the current manuscript. In fact, we did attempt a number of analyses to call TE insertion-site polymorphisms within genomes (as opposed to the simpler intergenomic analysis presented in Figure 4A), e.g., TEPID, TEFLoN, Jitterbug, McClintock, to name a few. Unfortunately, most of these analyses require both a “gold standard” reference genome and/or reference TE annotation in order to call TE genotypes, both of which are lacking in our dataset. Given this limitation, it was not possible to robustly assess the accuracy of the results of these analyses. Thus, we chose to focus on TE content evolution more generally, presenting analyses that are less sensitive to the genotype of a given TE insertion.

We agree that this would be very interesting to explore in future, particularly in the context of looking for signatures of recombination and selection. A thorough analysis of nucleotide diversity is a complex project that is beyond the scope of the current manuscript, but it is an area of ongoing research in our lab. A full analysis of intragenomic TE polymorphisms will be an exciting complementary project to revisit when better reference genomes for bdelloids become available.

16) Question related to demography and selection: would it be possible to obtain estimates of the effective population size for these clades? It would be interesting to have such an estimate to get an idea of the efficiency of purifying selection against TEs, and whether Muller's ratchet could explain the current abundance of TEs (in the case of moderate/small effective population sizes).

This sounds like an interesting idea, but we think it is beyond the scope of the current study. There are no estimates of effective population size for bdelloids except inferences from levels of variation for DNA sequence markers. It might be possible to compare within and between population variation for silent DNA substitutions versus TEs, but this would add quite a bit of complexity for a side analysis. When we are able to increase sampling of TEs within populations in future work, we will follow the reviewer’s suggestion to attempt to infer selection acting upon them.

17) We liked the idea of using the ABC to test for consistency with asexuality, but to what extent is it biased by non-constant transposition rates, which cannot be properly modeled by the coalescent simulation? One would also assume these simulations do not take into account past changes in demography? This is not necessarily a major issue, as long as these limitations are mentioned.

We have added a description of the assumptions of the ABC test to draw attention to limitations. We do not think it would be affected particularly by non-constant transposition rates. As now discussed in the text, the main limitation to the test is whether conditions permit sufficient polymorphism of TE presence/absence in multiple insertion sites to test for statistical associations between those different loci. That will depend on the usual factors affecting genetic variation (population size, average transposition rate) rather than non-constant transposition rates per se.